# Genomic data integration by WON-PARAFAC identifies interpretable factors for predicting drug-sensitivity in vivo

Yongsoo Kim[1,2,3], Tycho Bismeijer [2], Wilbert Zwart[1,4,6]*, Lodewyk F.A. Wessels[2,5,6]* & Daniel J. Vis[2,6]*

Integrative analyses that summarize and link molecular data to treatment sensitivity are crucial to capture the biological complexity which is essential to further precision medicine. We introduce Weighted Orthogonal Nonnegative parallel factor analysis (WON-PARAFAC), a data integration method that identifies sparse and interpretable factors. WON-PARAFAC summarizes the GDSC1000 cell line compendium in 130 factors. We interpret the factors based on their association with recurrent molecular alterations, pathway enrichment, cancer type, and drug-response. Crucially, the cell line derived factors capture the majority of the relevant biological variation in Patient-Derived Xenograft (PDX) models, strongly suggesting our factors capture invariant and generalizable aspects of cancer biology. Furthermore, drug response in cell lines is better and more consistently translated to PDXs using factor-based predictors as compared to raw feature-based predictors. WON-PARAFAC efficiently summarizes and integrates multiway high-dimensional genomic data and enhances translatability of drug response prediction from cell lines to patient-derived xenografts.

[1] Division of Oncogenomics, Oncode Institute, The Netherlands Cancer Institute, Amsterdam, The Netherlands. [2] Division of Molecular Carcinogenesis, Oncode Institute, The Netherlands Cancer Institute, Amsterdam, The Netherlands. [3] Department of Pathology, VU University Medical Center, Amsterdam, The Netherlands. [4] Department of Biomedical Engineering, Eindhoven University of Technology, Eindhoven, The Netherlands. [5] Faculty of EEMCS, Delft University of Technology, Delft, The Netherlands. [6] These authors jointly supervised this work: Wilbert Zwart, Lodewyk F.A. Wessels, Daniel J. Vis. *email: w.zwart@nki.nl; l.wessels@nki.nl; d.vis@nki.nl

Precision medicine aims to deliver the right drug to the right patient at the right time[1], which requires the ability to predict clinical benefit for anti-cancer drugs. To improve the understanding of which molecular alterations underpin drug sensitivity researchers require a large compendium of molecular profiles and drug response. The prime example of such resource is the Genomics of Drug Sensitivity in Cancer 1000 (GDSC1000) data[2], one of the largest and best-characterized cell lines screens publicly available. It holds data on mutations, DNA copy number, and gene expression of 1000 cancer cell lines comprising 55 tumor types tested for 265 compounds. The in-vitro models have enabled many clinically relevant discoveries[3]. Alternatively, in-vivo, patient-derived xenografts (PDXs)[4] have been established by transplanting human tumors into mice. The PDX encyclopedia (PDXE), comprising 1075 PDX models is a large resource for PDX molecular and drug response profiles. Similar to the GDSC1000, it contains data on mutations, DNA copy number, and gene expression and PDXs have been profiled for their pharmacological response to one of the 36 compounds/treatments[4].

Genomics data is high dimensional (many features) and consist of multiple data types (mutation, DNA copy number alteration, and gene expression), which makes it difficult to construct interpretable prediction models[5]. Sparse regression analysis, such as TANDEM that elegantly prioritizes interpretable features (i.e. select mutation features over gene expression), partially addresses this problem. However, sparse regression discards correlated gene features that can be relevant for interpretation, such as expression change of a gene accompanied by its mutation (i.e. cis-association). Instead, a-priori dimensionality reduction can in advance reduce the complexity of multi-type genomics data[6] (Supplementary Fig. 1A, B). More specifically, joint factorization approaches identify the correlation structure across the multiple data types, which has been used to identify clusters of features (joint NMF) or samples (JIVE and iCluster)[7–9]. However, this way of handling multiple matrices does not model the relationship between the variation of a gene across data types, such as copy number amplification and gene expression changes. Some other methods do appreciate these cis-associations (e.g. CON-EXIC[10] and iPAC[11]) but identify a sparse set of potential driver genes while ignoring the remaining genes. The alternative is to organize the data in a sample by gene by data type cube, which preserves the relations between the different data types and the samples. One of the better-known methods for factorizing multi-way data is parallel factor analysis (PARAFAC[12]; Supplementary Fig. 1C), which has not been applied to genomics data integration.

WON-PARAFAC, which is introduced here, is an integrative framework based on PARAFAC with the following three constraints. First, a weighting scheme at the data-type level ensures that data types with large variance (such as gene expression) do not dominate the analysis. Second, an orthogonality constraint is introduced to decrease the correlation between factors. Third, non-negativity is enforced to obtain sparser solutions that are much easier to interpret[13,14]. We named the new method Weighted Orthogonal Non-negative (WON)-PARAFAC.

With WON-PARAFAC, we identify cancer factors from in-vitro pan-cancer cell line data (GDSC1000) and we demonstrate that these translate reliably to the in vivo setting (PDXE project). The cell-line-derived factors capture characteristics specific to a tissue type in both cell lines and PDXs and are more consistently predictive for treatment response in both model systems as compared to raw features. Taken together, WON-PARAFAC offers a new level of data integration that appreciates the links between samples and features and data types, and provides interpretable results while allowing for improved translation of in-vitro drug sensitivity to animal models of multiple cancer types.

## Results

**Deriving factors from cell line data using WON-PARAFAC.**
We obtained mutation (MT), copy number (CN), and gene expression (GE) data from the largest cell line screen, the GDSC1000[2]. Since we focus on cancer, we selected the largest cancer-specific gene-panel we could find, which was the Center for Personalized Cancer Treatment (CPCT, The Netherlands) mini cancer genome panel[15] comprising 1977 genes of which 1815 (92%) were available in the cell line panel[16]. Mixed sign data (i.e. gene expression and copy number changes) were separated on sign, followed by taking absolute values to ensure non-negativity. These data splits resulted in five data type matrices, referred to as GE(+) and GE(−) (from GE), CN(+) and CN(−) (from CN) and MT (Fig. 1a). We organized the data as a three-way dataset (a data cube) of 1815 genes by 935 cell lines by five data types, with the same ordering of genes and cell lines in each data type (Fig. 1b). WON-PARAFAC decomposed the cube into three sparse matrices with 130 factors (Fig. 1c, S2, and S3; see Methods), where each factor has a gene-factor, a cell-factor, and a data type (DT)-factor.

**Constraints enhance data integration and interpretation.**
WON-PARAFAC combines the non-negativity constraint with a data type weighting scheme and a gene mode orthogonality constraint. The weighting scheme standardizes the relative importance of each data type and allows for better incorporation of MT and CN (Supplementary Fig. 4A–C). The orthogonality constraint reduces correlation among gene-factors. It also improved identification cis-alterations, in which one gene is altered in two or more data types (e.g. copy number gain and higher gene expression)[17]. Without the weighting and orthogonality constraints, the factorization fails to properly perform integration between data types, especially between GE(−) and MT data (Supplementary Fig. 4D). In line with previous studies, DT-factors in WON-PARAFAC identified that copy number alterations are strongly associated with gene expression data, resulting in a cosine similarity of $c = 0.31$ between CN(+) and GE(+) and $c = 0.38$ between CN(−) and GE(−)[18].

Each gene, cell line, and DT-factor can be interpreted using their loadings (coefficients). For instance, Factor 41 has (1) large loadings in the DT-factor for GE(−), MT and also CN(−) indicating that these data types contribute most to this factor (Fig. 1d, bottom-left) and (2) large loadings for CDKN2A and CDKN2B in the gene factor, indicating that this factor captures the variation in these genes (Fig. 1d, top-left). The top cell lines in the cell-factor had co-alterations of CDKN2A and CDKN2B in all three data types (Fig. 1d, right). Of note, multiple data types contribute to almost half (58) of the factors (Supplementary Fig. 5). These factors capture sets of genes potentially exhibiting cis-effects, interesting examples include (1) Factor 94 which captures the co-amplification and overexpression of the MYC oncogene along with its proximal genes (ASAP1, PTK2; Supplementary Fig. 6); and (2) Factor 58 which captures the mutations and decreased expression levels of PTEN, RB1, and TP53 all of which are tumor suppressors (Supplementary Fig. 7).

**Interpretation of the factors.** We further interpreted factors by relating them to tissue type, pathways, and treatment response (Fig. 2). For each cell-factor, we performed a cell set enrichment analysis (CSEA, similar to gene set enrichment analysis (GSEA)[19]) to link cancer types to each cell line-factor (Fig. 2a, e). In parallel, an unbiased GSEA was performed on the coefficients obtained by

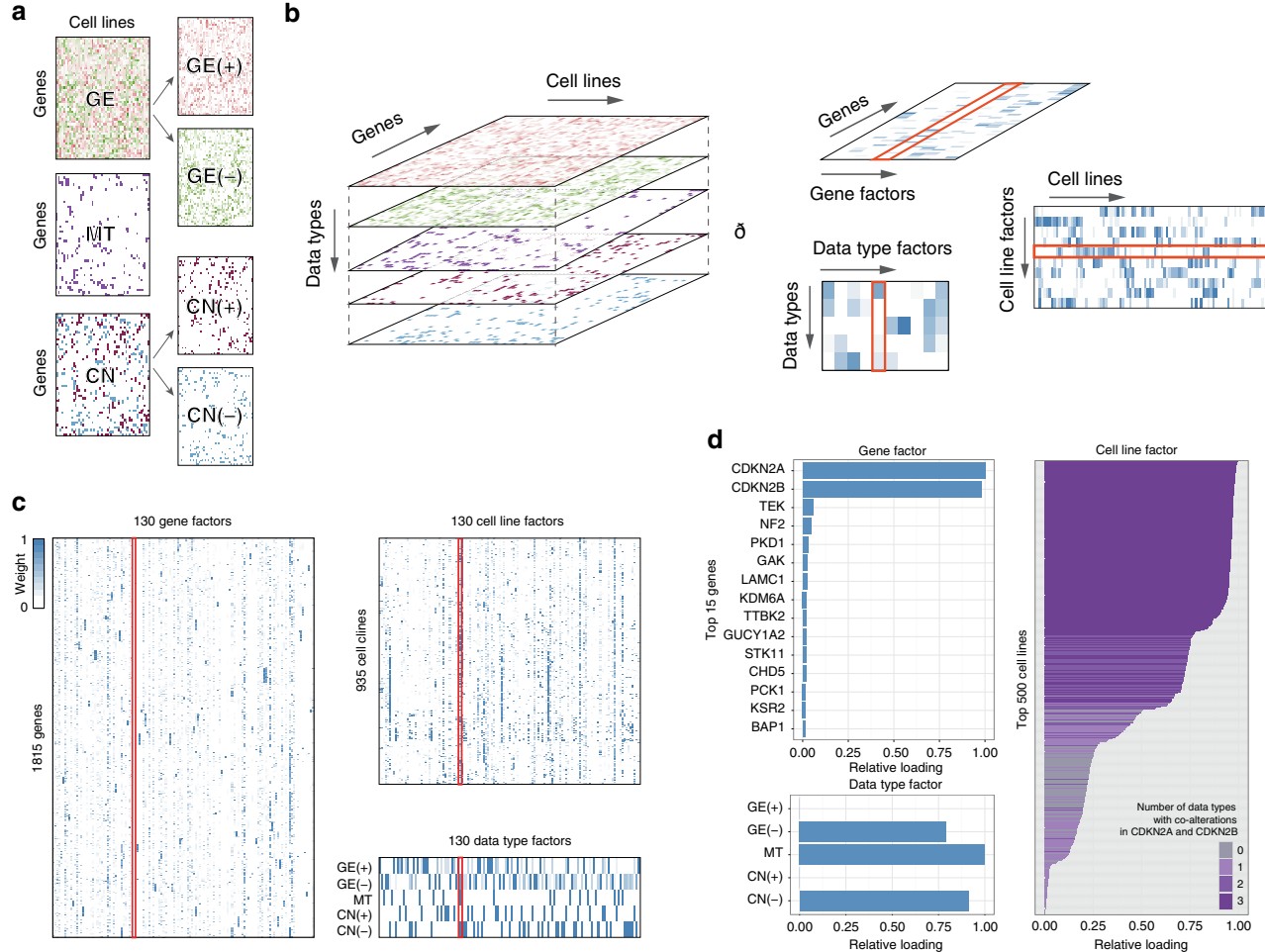

**Fig. 1** The framework of WON-PARAFAC. **a** Individual data types represented in genes-by-cell-lines matrices with matched gene and cell line indices across the data types in which mixed sign data (i.e. gene expression and copy number) are split into two positive matrices. **b** Data cube formed by stacking the five matrices, where the modes span genes, cell lines, and data types. WON- PARAFAC produces three sparse factor matrixes for each of the modes: gene factor (gene-factor), cell line factor (cell-factor) and data type factor (DT-factor) matrix. **c** The three sparse factor matrices from WON-PARAFAC as heatmaps, with the 41st factor highlighted. **d** 41st factor loadings of top 15 genes (gene-factor; top left), data types (DT-factor; bottom left), and top 500 cell lines (cell-factor; right). Color of bars in the bar plot for cell-factor indicates the number of data types with CDKN2A and CDKN2B altered in each cell line

regressing the global gene expression data on the cell-factors. We used the KEGG pathways as well as biological processes and hallmark gene sets from MsigDB[20] (v5.2) to identify enriched pathways in the 130 cell-factors (Fig. 2a, f). Finally, we employed elastic net (EN) regression to link the factors to drug response. EN was chosen, as it achieves similar predictive performance as other machine learning methods[21]. EN models were trained on the 130 cell-factors to predict the drug sensitivity (quantified by area under the dose-response curve, or AUC) for each of the 265 compounds in the cell line panel (Fig. 2a, g).

We illustrate the factor interpretation with a selected set of factors (Fig. 2b–d), and their associations with tissue type, pathways and drug response data (Fig. 2e–g). Factor 41 is associated with CDKN2A/B and with mesothelioma, kidney, and glioma (Fig. 2e), where the loss of CDKN2A/B is common (Fig. 2b and Supplementary Fig. 8A). At the same time, Factor 41 gene loadings are associated with down-regulation of interferon-alpha (IFa) response genes (Fig. 2f), which is worth noting as IFa is a treatment option in mesothelioma[22]. On the drug-association level, Factor 41 associates to CDK4/6 inhibitor *Palbociclib* (PD-0332991), for which CDKN2A/B loss is a sensitivity biomarker[23]

(Fig. 2g). For Factor 12 which is associated with breast cancer (Fig. 2e), the same integrative analysis reveals higher ERBB2 and ESR1 expression (Fig. 2b and Supplementary Fig. 8B) coupled with, amongst other, ERBB/MAPK/mTOR/Notch signaling (Fig. 2f) and shows sensitivity for Afatinib (BIBW2992), which targets ERBB2 (Fig. 2g).

Globally, we found expected grouping of cancer types (e.g. hematological cancers) but also unexpected combinations (e.g. mesothelioma and glioma share 4 factors commonly enriched; Supplementary Figs. 9–11). Among gene sets used in GSEA, we found frequent up-regulation of cancer pathways across factors, including MAPK, ERBB-family, and insulin signaling pathways (Supplementary Fig. 12). At the same time, mitochondria-related pathways such as oxidative phosphorylation and mitochondrial translation were frequently downregulated, which is in line with cancer cell's reliance on aerobic glycolysis instead of oxidative phosphorylation for ATP production[24].

**Treatment response prediction based on the factors.** We trained EN models on the 130 cell-line factors (predictors) and the area under the dose-response curve (AUC) as the measure of drug

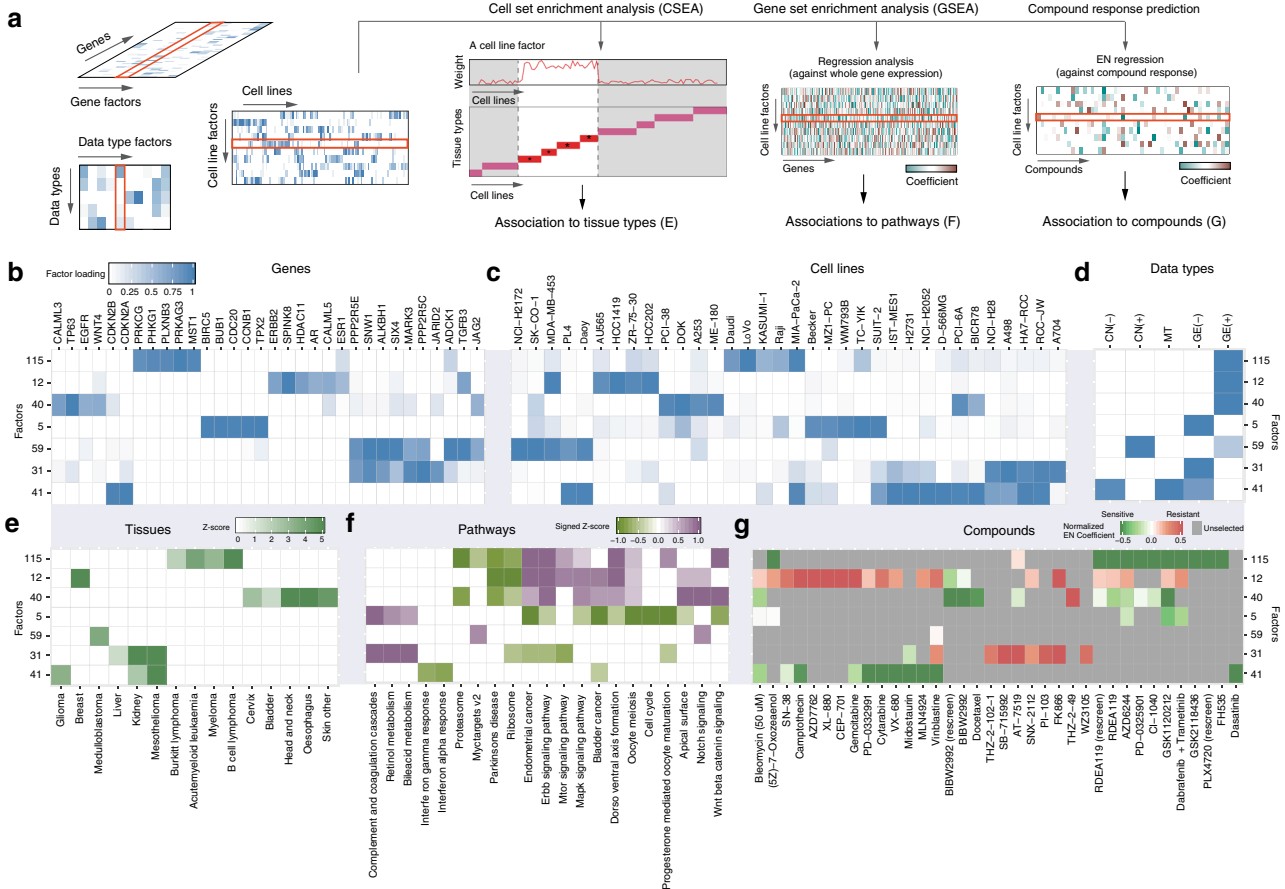

**Fig. 2** Interpretation of factors obtained from WON-PARAFAC. **a** Assessment of tissue types enrichment; gene set enrichment and association to drug responses for cell-factors. **b–d** (**b**) Gene factor (gene-factor) matrix, **c** cell line factor (cell-factor) matrix, and **d** data type factor (DT-factor) matrix visualized in heatmaps, for the selected factors and their top associated genes, cell lines, and data types, respectively. Factor loading is indicated by color gradient, which is the strength of the association. **e–g** Associations between the selected factors and tissue types (**e**), enriched pathways (**f**), and compounds (**g**) in heatmaps where their color gradient represents the significance of association in Z-scores, signed Z-scores, and coefficients from EN, respectively. The sign in Z-score indicates up or down-regulation of pathways, and sign in EN coefficients indicates sensitivity (negative) or resistance (positive) to compounds

sensitivity (outcome) using nested 10-fold cross-validation and repeated this for every drug. In parallel, we trained EN models on all (n = 5445) features and denote this as the 'raw EN'. To contrast the performance of WON-PARAFAC and the raw EN reference model to a state of the art approach, we also performed the analysis using TANDEM, a two-stage EN approach for improved data integration and interpretation[5]. The nested cross-validated performance was similar between the three approaches, with the average correlation of the predicted and actual responses across all compounds being r = 0.19 for factor-based EN, r = 0.22 for the raw EN reference and r = 0.23 for TANDEM (Fig. 3a, and Supplementary Fig. 13A). The performance across all compounds of factor-based ENs was highly correlated with that of raw EN reference models (r = 0.93, p < 2.2e-16; Pearson correlation test) and TANDEM (r = 0.98, p < 2.2e-16; Pearson correlation test). TANDEM significantly improved the contribution of mutation and copy number data overall data predictor models, as reported in the paper[5]. WON-PARAFAC was able to improve on the copy number contribution by 0.12 at no apparent cost to predictive performance (Supplementary Fig. 13B). This indicates that our 130 factors contain sufficient information to explain the drug response with a 42-fold (5445/130) feature reduction. The drugs better predicted with raw features tended to rely on the features less-well reconstructed by the factors, which typically are

mutations due to the low event rates (Fig. 3b and Supplementary Fig. 14).

We can interpret the contribution of factors to predictions of drug response based on the sign and size of the EN model parameters. For example, factor-based EN predicts response to Afatinib (inhibits EGFR and ERBB2), using four cell line factors (Fig. 3c). First, we note that the contribution of these factors to the prediction correlates with the absolute EN coefficient (Fig. 3c). Second, all these factors have negative EN coefficients implying that a high value of the factor corresponds to a low response activity area, i.e. sensitivity to the drug (Fig. 3c, top). This is in agreement with that fact that Factor 40 is associated with high EGFR expression (Fig. 2b), which inhibited by Afatinib (Fig. 2g). Among the chemotherapeutic drugs, TOP1-poisons Camptothecin and Irinotecan (SN-38) are well-predicted (r = 0.37 and r = 0.34, respectively; Fig. 3a, b). Factor-based ENs selects 20 and 9 factors to predict the response to Camptothecin (Fig. 3d) and Irinotecan (Fig. 3e), respectively, of which we selected the union of the top five factors with the largest absolute EN coefficients from two drugs (in total six factors; darker shaded bars). Together, the top six factors significantly separated cell lines on response to both Comptothecin/Irinotecan (Fig. 3f), indicating the ENs identi-fied sensitivity and resistance factors.

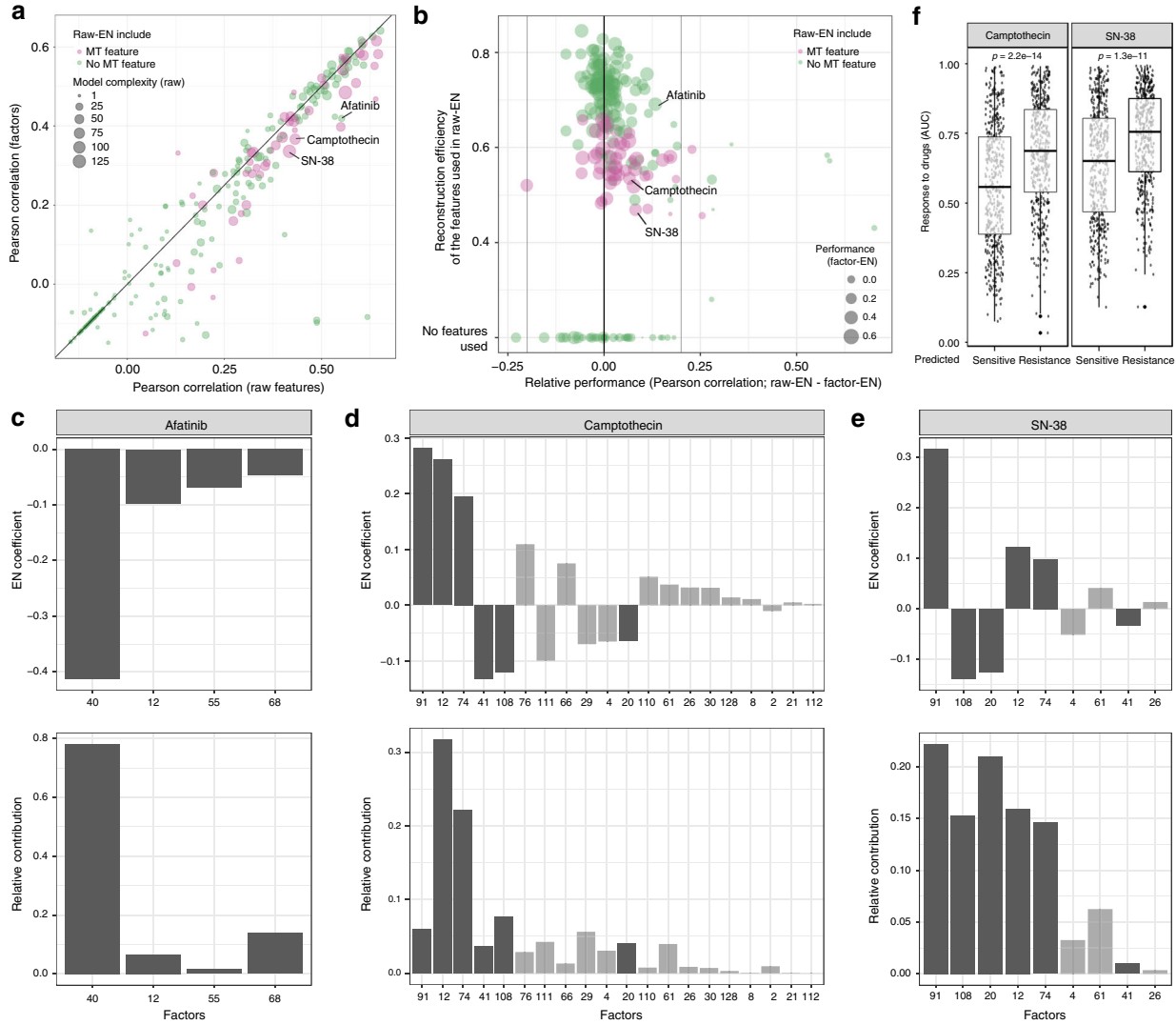

**Fig. 3** Associations between factors with tissue type, pathways and drug responses. **a** Predictive performances of ENs trained with raw features (*x*-axis) and factors (*y*-axis) for 265 compounds in Pearson correlation. Usage of mutation data and the total number of features selected in the raw feature-based are indicated by node color and size. **b** The relative prediction performance of ENs trained with raw features (raw-EN) and factors (factor-EN; *x*-axis), compared with the reconstruction efficiency of the features in raw-EN model (average explained variation of the selected features by WON-PARAFAC; *y*-axis) in a scatter plot. Node color/size indicates the use of mutation data by raw-EN and performance of factor-EN, respectively. **c–e** EN coefficient (top; *y*-axis) and relative contribution (bottom; *y*-axis) of the factors (*x*-axis) predicts response to *Afatinib* (**c**), *Camptothecin* (**d**) and SN-38 (*Irinotecan*; **e**). The dark bar indicates the six factors used in the downstream analysis. For Camptothecin and SN-38, we selected the union of the top five factors with the largest absolute EN coefficients from in the two EN models. **f** Comparison of response to *Camptothecin* (left panel) and SN-38 (*Irinotecan*; right panel) between cell lines with higher loadings on sensitivity (predicted sensitive; left boxplot) and resistance factors (predicted insensitive; right boxplot), respectively. Standard notations are used for elements of the boxplot (i.e. upper/lower hinges: 75th/25th percentiles; inner-segment: median; and upper/lower whiskers: extension of the hinges to the largest/smallest value at most 1.5 times of interquartile range)

**Linking biology to treatment response through Networks**. Using the factors identified in the response prediction and their top enriched functions and tissue types (FDR < 0.2 from GSEA and CSEA, respectively), we further explored the underlying biology. For *Afatinib*, there were four factors associated with sensitivity: Factors 12, 55, 40, and 68 (Fig. 4a). The factors frequently associated with active ERBB pathways. Among them is Factor 12, a factor mostly driven by overexpression of ERBB2 itself. Notably, Notch-signaling is also associated with the same factors as the ERBB pathway, suggesting crosstalk between the pathways[25]. The cancer types strongly associated with these factors were enriched for the cell lines sensitive to Afatinib, such as head and neck cancer (12 out of 29 lines with AUC < 0.6,

including HSC-3 and CAL-27; Fig. 4b). Note that, rather than EGFR mutation—the clinically accepted biomarker for Afatinib sensitivity[26]—EGFR expression is selected instead. This discrepancy may be due to the less efficient reconstruction of mutation data.

Figure 4c shows the top six factors (factors with the largest coefficients from EN) associated with response to the TOP1-poisons Camptothecin and Irinotecan. The factors that predict sensitivity to these drugs are associated with activation of the Intrinsic apoptotic pathway by p53 class mediator (Factor 20), reduced interferon-alpha and gamma response (Factor 41), and T-cell receptor signaling pathway (Factor 108). It has been shown that apoptosis induced by Camptothecin is partially dependent on

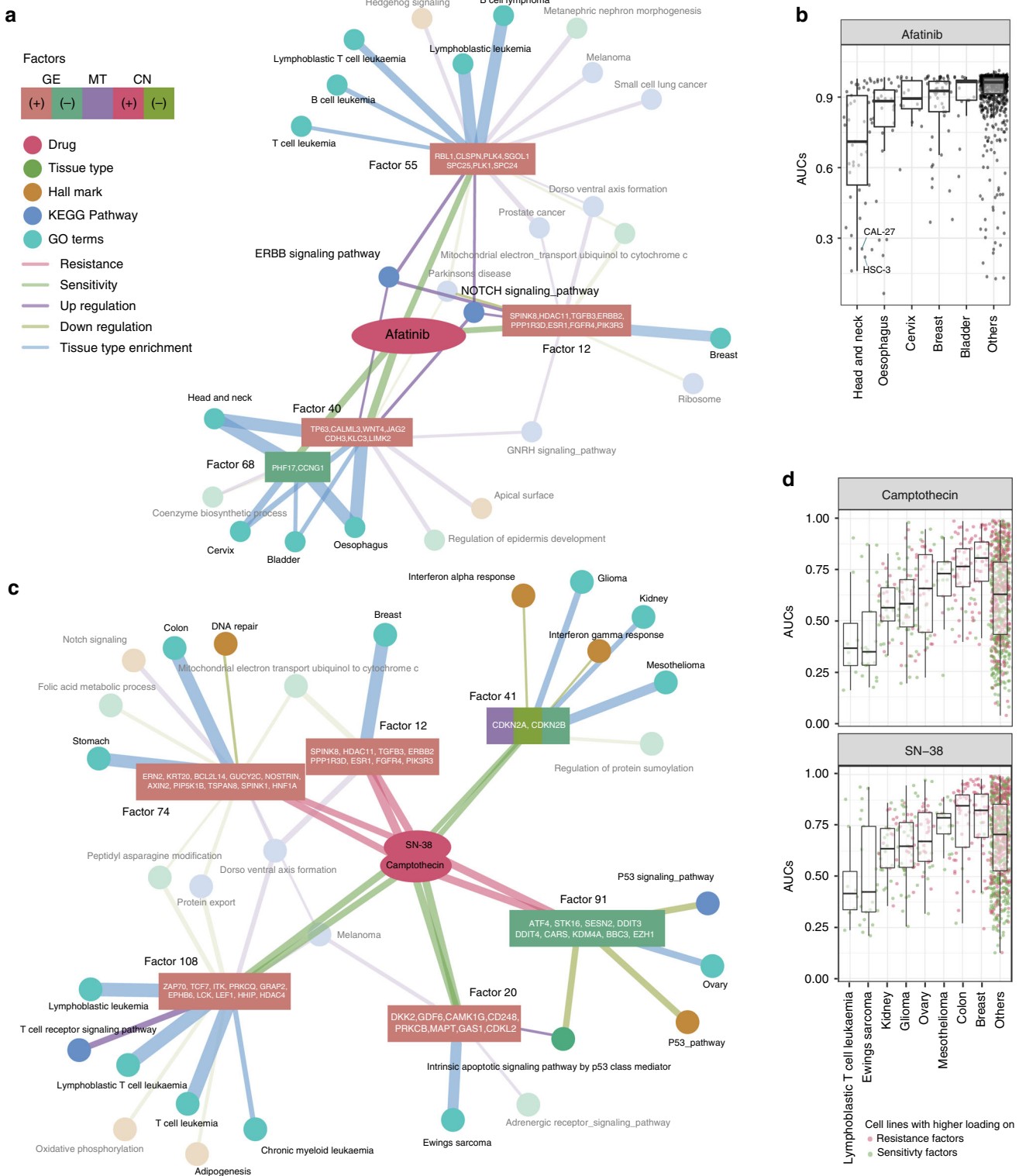

**Fig. 4** Network-based combined interpretation of factors that explain drug responses. **a** Top factors (square nodes) explaining the response to *Afatinib* (BIBW2992; pink node) with their associated tissue types (green nodes) and functions (dark yellow, blue and blue-green nodes) in a network. Listed in the corresponding node are the top genes from each factor indicating the factor. The types of node and edges are denoted by colors indicated in the figure. The nodes mentioned in the text are highlighted. **b** *Afatinib* response of cell lines by tissue types highlighted in **a**. Two sensitive head and neck cancer cell lines (CAL-27 and HSC-3) are indicated. Standard notations are used for elements of the boxplot (i.e. upper/lower hinges: 75th/25th percentiles; inner-segment: median; and upper/lower whiskers: extension of the hinges to the largest/smallest value at most 1.5 times of interquartile range). **c** Top factors that explain the response to TOP1 inhibitors, SN-38 (*Irinotecan*) and Camptothecin represented in a network (format as in **a**). **d** Camptothecin (top panel) and SN-38 (*Irinotecan*; bottom panel) response of cell lines by tissue types highlighted in **c**. Color of nodes indicates higher loading in sensitive factors (green) and resistance factors (red) for each cell line. Standard notations are used for elements of the boxplots

TP53[27], and cell lines with acquired resistance to interferon are also reported to be more resistant to Camptothecin[28]. Finally, TCR signaling pathway is a T-cell-specific pathway, and Factor 108 is also enriched with T-cell-driven tumor types (for example, T-cell leukemia), which are the most sensitive to the TOP1-poisons (Fig. 4d). This is also consistent with high clinical response rate of T-cell leukemia to irinotecan[29].

Similarly, resistance to TOP1-poisons associated with repression in the P53 pathway (Factor 91) and down-regulation of DNA repair pathway (Factor 74). Factor 12 also predicts insensitivity to TOP1-poisons and includes the canonical oncogenes such as ERBB2, TGFB3, ESR1 (AR), and FGFR4 that are strongly associated with breast cancer. Data for TOP1-poisons in breast cancer is limited. Phase II trials in refractory patients have shown response rates similar to anthracyclines (30%) and taxanes (12%) versus 5–25% for irinotecan[30], without stratifying patients. Our factor 12 points to tumors that are driven by ERBB-family members, growth factors, and hormone receptors (ESR1/AR). Note that Camptothecin and its analog, Irinotecan, and Topotecan, have been approved for colon[31] and ovarian cancer[32] while colon and ovarian cell lines are the most insensitive to these drugs in GDSC1000 (Fig. 4d). The clinical response rate in colorectal cancer is similar to that in breast (16–27%[31]) and our results point to subpopulations more likely to respond unfavorably which could inspire follow up experiments. Cumulatively, the factors allow us to functionally interpret the drug sensitivity phenotype and pinpoint to potential mechanisms of response.

**Invariance of the factors in patient-derived xenograft data.** WON-PARAFAC compresses genomic multi-dimensional data in a concise set of interpretable factors while retains predictive ability. We wondered if the factors are generalizable features that allow for translation of drug response data from cell lines towards in vivo models. To test this, we employed the PDX-encyclopedia. Among all PDX models, molecular profiles of 399 models are available. We fixed the cell line-derived gene-factor and DT-factor, which represents the latent biological structures in cell lines, and computed the 130 PDX factors. The resulting factors fit the molecular PDX data at a level comparable to the cell line data (44% vs 51% variation explained), substantially higher than sample-feature permuted data (Fig. 5a). Among the cancer types, pancreatic ductal carcinoma (PDAC; $p = 0.049$; $t$-test) and colorectal carcinoma (CRC; $p = 0.0035$; $t$-test) were comparably reconstructed between PDX and cell lines, as compared to skin cutaneous melanoma (SKCM; $p = < 0.0001$; $t$-test), breast cancer (BRCA; $p < 0.0001$; $t$-test), and non-small cell lung cancer (NSCLC; $p < 0.0001$; $t$-test; Fig. 5b). Cancer-type associated factors, derived from the cell lines, show a strong association with PDXs of the same cancer type. BRCA and SKCM associated factors are very specific for the cancer type, whereas for CRC, NSCLC, and PDAC, there is some cross-cancer type association for the factors, in both cell lines and PDX (Fig. 5c). The cross-cancer type association hints at a common biological factor, for example, Factor 57 that is associated with both CRC and PDAC, suggesting a link with (the cancer hallmark) oxidative phosphorylation (Supplementary Data 1; sheets 4 and 5) Furthermore, the cell lines and PDX of the same cancer types are more similar in the factor-based representation. In $t$-SNE[33] spaces derived from either factors or raw features (Supplementary Figs. 15 and 16), cell lines and PDXs are generally closer in factor space except for CRC for which the similarity between cell lines and PDXs is comparable between the factor and raw spaces (quantified using Fisher criterion; Fig. 5d). This suggests that invariant patterns between the two model systems are better captured by the factors.

Next, we investigated whether a predictor of drug response trained on the cell lines would correctly predict sensitivity in PDX models. To this end, we trained for each drug an EN model to predict the response (measured by AUC) on the cell lines from the 130 cell-factors. We applied the ENs to the PDX-factors and compared the best average response, the sensitivity readout of the PDXs, with predicted AUCs of the same (pharmacological class of) compounds (Supplementary Figs. 17 and 18). The predicted AUCs were positively correlated with the best average response for the majority of compounds. The association was statistically significant for Encorafenib, Trastuzumab, and Binimetinib in a subset of tissue types (Fig. 5e). More specifically, the PDX response to Binimetinib can be predicted in CRC and NSCLC, but not in BRCA, PDAC, and SKCM PDXs, even though response to Binimetinib could successfully be predicted in cell lines. The PDX result is in line with clinical observations that MAPK inhibition failed to show benefit in PDAC[34] and melanoma[35], while clinical trials for NSCLC (NCT03170206) and CRC (NCT02928224) are still ongoing.

For comparison, we compared factor-based ENs with raw feature-based ENs by prediction performance in the PDXs. Raw feature-based models were significantly predictive for only Erlotinib, an EGFR targeting drug (Fig. 5e). Furthermore, predictive performance significantly dropped in PDXs for raw feature-based ENs ($p = 0.0024$; $t$-test), indicating superior transferability of factor-based models (Fig. 5f). In particular, for Afatinib an EN trained on raw cell line features ($r = 0.57$, $p < 0.001$; Pearson correlation test) outperforms the EN trained on the factors in cell lines ($r = 0.42$, $p < 0.001$). However, the raw feature-based EN failed to predict response to trastuzumab; the closest match of Afatinib in PDXs ($r = -0.11$, $p = 0.53$; Pearson correlation test). In contrast, the factor-based EN performed well on both cell lines ($r = 0.42$, $p < 0.001$; Pearson correlation test) and PDXs ($r = 0.4$, $p = 0.014$; Pearson correlation test; Fig. 5e, g). The EN trained on raw features picks up ERBB2 expression as a predictive feature, which is indeed correlated with the response in PDXs (Supplementary Fig. 19). However, the other features used by the model deteriorate the performance. The PDX-response predictions based on raw features underperformed relative to the factors (Supplementary Fig. 20). We attribute the enhanced predictive performance to the data integration by WON-PARAFAC resulting in more stable predictors with fewer outliers. The compressed representation obtained by factorization aid the interpretation of the data while it allows for state-of-the-art predictive performance, and most importantly improves drug response translation from in vitro to in vivo.

## Discussion

WON-PARAFAC provides a compact summary of integrated molecular/omics data. The method provides results that are amenable to easy interpretation, and above all, it carries predictive capacity in a translational setting. This seamless amalgamation of three major topics in oncology; data integration, interpretation, and translation in a single method allows for simultaneous linking of alterations across data types that are preserved in multiple samples. We show that naïve integration by concatenating omics data sets, here illustrated with EN, is more exposed to outlier values and platform/species differences than the WON-PARAFAC approach. Furthermore, projecting cell lines and the PDX data into 130-factor space yields a homogeneous representation of PDXs and cell lines, with comparable reconstruction rates for both of them given the in-vitro/in-vivo differences, without any explicit optimization towards that end. This means that the 130 factors capture generalizable molecular characteristics of cancer that are preserved between cell lines and

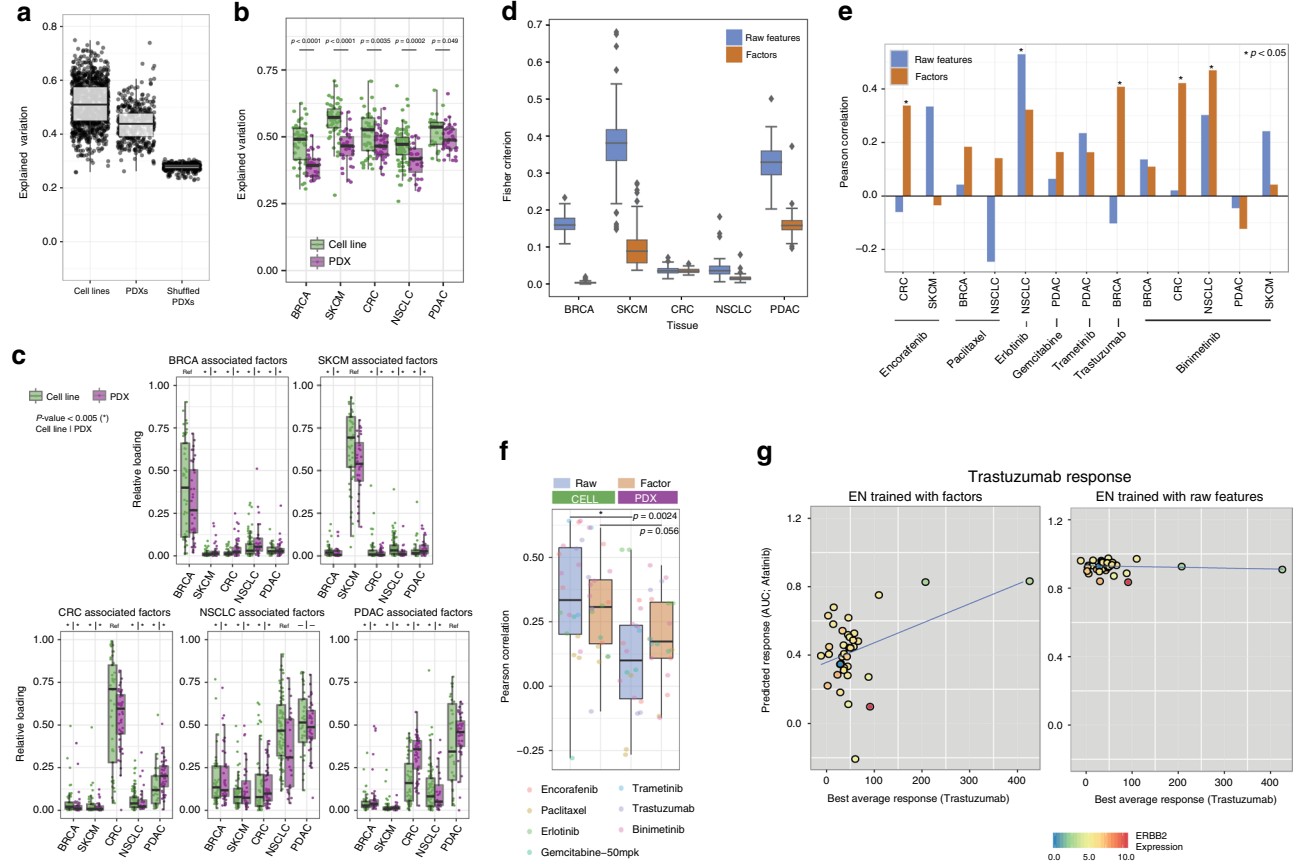

**Fig. 5** Application of trained model using pan-cancer cell line data to pan-cancer PDX data. **a** Explained variation of cell lines, PDXs and permuted PDXs data using cell line-driven gene-factors and DT-factors in boxplots where each point indicates each sample. Standard notations are used for elements of the boxplot (i.e. upper/lower hinges: 75th/25th percentiles; inner-segment: median; and upper/lower whiskers: extension of the hinges to the largest/smallest value at most 1.5 times of interquartile range; also for **b**, **c** and **f**). **b** Explained variation of cell lines and PDXs for each of the five tissue types: breast cancer (BRCA), skin cutaneous melanoma (SKCM), colorectal cancer (CRC), non-small cell lung cancer (NSCLC), and pancreatic cancer (PDAC). *P*-value indicates difference in mean between cell line and PDX. **c** Relative factor loadings for cell lines and PDXs by factors associated with the cancer type. The significant difference in loading relative to the reference tissue type ($p < 0.005$; *t*-test) is indicated with an asterisk at the top of each boxplot. **d** The separation between cell lines and PDXs per tissue type in 100-repeated *t*-SNE embedding measured by the Fisher criterion in boxplots, stratified by the five tissue types. Blue and orange boxes indicate raw features and factors. Standard notations are used for elements of the boxplot. **e** Correlation between best average response and predicted drug responses (AUC) using ENs based on raw features and factors, represented in a bar plot. Asterisks indicate drugs with significant predictive performance ($p < 0.05$; Pearson correlation test). **f** Prediction performance of drugs per tissue type in cell lines (green) and PDXs (purple) using raw (blue) and factors (orange) in a boxplot. Individual color-coded dots represent the compounds used in the analysis. *P*-values comparing the performance between cell lines and PDXs for each feature are indicated on the top. **g** Predicted (*x*-axis) and measured (*y*-axis) response of PDXs to *trastuzumab* for ENs based on factors (left scatter plot) and raw features (right scatter plot), where each dot is a PDX. Color in the dots indicates expression level the drug target ERBB2 in PDXs. The blue line is the linear regression line

PDX models. Based on this, we conclude that EN models using our factors perform more consistently between model systems.

To the best of our knowledge, this is the first example of a method to handle a data cube of integrated genomic datasets in which associations between different alterations of the same genes (i.e. cis-effect) are appreciated. Due to the structural constraint in WON-PARAFAC, these cis-relationships (for example, a mutation associated with copy number loss of the same gene) were captured. Furthermore, trans-effect of factors is also captured by regressing the factors onto complete gene expression data, which was used in GSEA. Conventional two-way matrix-based approaches, such as iCluster[9] and IntNMF[36] (Supplementary Fig. 1A), are less efficient in capturing the cis-associations and also lack systematic interpretations including association to drug sensitivity.

Our study is bound by the limitations of the model systems used. The GDSC1000 is the largest extensively characterized cell line panel currently available that is screened for differential compound sensitivity. However, the number of cell lines in any cancer type subgroup is still limited and biased. For example, 50 cell lines represent colorectal cancer (CRC) which covers the majority of colorectal cell lines in existence globally, but it still under-appreciates the complexity of the tumor type. The low sample size and the implicit selection bias in generating cell lines pose a challenge for predicting response in a translational setting. The same limitation holds for PDX models; In vivo models that capture stromal interactions, 3D structure, and drug metabolism. Furthermore, the host mice are immunocompromised to prevent host-graft rejection of the human tumor tissue.

Finally, we reason that drug-repurposing designs could benefit from the compressing raw features concept proposed here using a factorization approach, as we speculate that the superior translational ability might also hold true for translating in vitro screens to human trials. While most current repurposing studies

(including the *Drug Rediscovery Protocol-Trial* (NCT02925234) and the American Society of Clinical Oncology's Targeted Agent and Profiling Utilization Registry (ASCO-TAPUR) study (NCT02693535)) focus solely on mutation and copy number data, the factors may identify specific gene expression patterns with predictive value. The clinical utility of gene expression data has been highlighted before, and our 130 cancer factors add to that argument. Furthermore, the factors capture a common molecular basis between tissue types (Fig. 3a) and carry the same degree of tissue specificity in vitro as in vivo (Fig. 5c). The associations with drug sensitivity, inferred by EN, can be used to identify which treatment that is already registered for one cancer type can also be used in other cancer types. We illustrated a potential case of drug repurposing for Afatinib and Irinotecan/ Camptothecin (see Fig. 4a/c).

In summary, we have demonstrated that WON-PARAFAC efficiently compresses high dimensional multi-genomics data from in vitro cell lines into an interpretable yet invariant representation that also applies to in vivo PDX data. It allows for sensitivity prediction in cell lines to anti-cancer compounds, but more importantly, also allows for the sensitivity prediction in PDXs using a structure that was trained exclusively on cell lines. The complex biology in cell lines is both robust and relevant for predicting drug response in an in vivo model system. We have shown that the underlying biology can be disentangled, interpreted, and are thus allowing for hypothesis generation and prediction beyond what would be achievable without data integration.

## Methods

**GDSC1000 pan-cancer cell line data**. Processed gene expression, copy number, and mutation data together with drug response measures (AUC) were obtained from the GDSC1000 webpage (http://www.cancerrxgene.org/gdsc1000/ GDSC1000_WebResources/Home.html). We standardized gene expression levels by mean centering and scaling with the standard deviation. The copy number data is reduced to the following three states: (1) 1 if copy number >5; (2) −1 if copy number ≤1; and (3) 0 otherwise. The thresholds for the copy number is consistent with that used in PDXE data (5 and 0.8 are used, respectively). The mutation data is also converted to binary. We obtained a subset of genes included in the Center for Personalized Cancer Treatment (CPCT, The Netherlands) mini cancer genome panel (Hoogstraat, Vermaat et al.[15]), which result in 1815 genes. We represented each of the data types by an 1815 gene by 935 cell line matrix with the consistent order of genes and cell lines.

**Weighted orthogonal non-negative PARAFAC (WON-PARAFAC)**. Non-negative PARAFAC is a combination of non-negative matrix factorization (NMF) and parallel factor analysis (PARAFAC)[12]. Specifically, in case of a $g \times c \times d$ data cube $\mathbf{X}$ spans $g$ genes, $c$ cell lines, and $d$ data types, the objective functions of non-negative PARAFAC with $k$ factors can be formulated as a collection of the following three NMF objective functions:

$$\min_{G \geq 0} ||\mathbf{Y}_{CD}\mathbf{G}^T - \mathbf{X}_{(G)}||_F^2, \min_{C \geq 0} ||\mathbf{Y}_{GD}\mathbf{C}^T - \mathbf{X}_{(C)}||_F^2, \min_{D \geq 0} ||\mathbf{Y}_{GC}\mathbf{D}^T - \mathbf{X}_{(D)}||_F^2.$$

(1)

The notations in the above equations are: (1) $\mathbf{G}$, $\mathbf{C}$, and $\mathbf{D}$ are $g \times k$, $c \times k$, and $d \times k$ matrixes that contain $k$ gene factors (gene-factors), $k$ cell line factors (cell-factors), and $k$ data type factors (DT-factors), respectively; (2) $\mathbf{X}_{(G)}$, $\mathbf{X}_{(C)}$, and $\mathbf{X}_{(D)}$ are $cd \times g$, $gd \times c$, and $gc \times d$, matrices that are unfolded from the cube $\mathbf{X}$ across genes, cell lines, and data types, respectively; and (3) $\mathbf{Y}_{CD}$, $\mathbf{Y}_{GD}$, and $\mathbf{Y}_{GC}$ are $cd \times k$, $gd \times k$, and $gc \times k$ matrices obtained from the Khatri-Rao product between $\mathbf{C}$ and $\mathbf{D}$, between $\mathbf{G}$ and $\mathbf{D}$, and between $\mathbf{G}$ and $\mathbf{C}$, respectively (for more details of the formulation, see Kim et al.[37]). By alternating least-squares optimization of the above objective functions, we can find the factor matrices $\mathbf{G}$, $\mathbf{C}$, and $\mathbf{D}$ that best approximates the original data cube $\mathbf{X}$. We can derive a multiplicative update rule by decomposing gradients to positive and negative parts. Suppose we can decompose a gradient of an objective function $E$ as follows:

$$\Delta E = [\Delta E]^+ - [\Delta E]^-,$$

(2)

where $[\Delta E]^+ > 0$ and $[\Delta E]^- > 0$. Then, multiplicative update for a parameter $\Theta$ is:

$$\Theta = \Theta \odot \frac{[\Delta E]^+}{[\Delta E]^-}$$

(3)

where $X \odot Y$ is the Hadamard product (element-wise product), $\mathbf{X}/\mathbf{Y}$ is the element-

wise division, and $(.)^\eta$ is the elementwise power. Note that $\eta$ is a learning rate ($0 < \eta \leq 1$), which we set to 1. Taking the same manner, multiplicative update rules of the parameters $\mathbf{G}$, $\mathbf{C}$, and $\mathbf{D}$ can be derived as follows:

$$\mathbf{G} = \mathbf{G} \odot \frac{\mathbf{X}_{(G)}^T \mathbf{Y}_{CD}}{\mathbf{G}\mathbf{Y}_{CD}^T \mathbf{Y}_{CD}},$$

(4)

$$\mathbf{C} = \mathbf{C} \odot \frac{\mathbf{X}_{(C)}^T \mathbf{Y}_{GD}}{\mathbf{C}\mathbf{Y}_{GD}^T \mathbf{Y}_{GD}},$$

(5)

$$\mathbf{D} = \mathbf{D} \odot \frac{\mathbf{X}_{(D)}^T \mathbf{Y}_{GC}}{\mathbf{D}\mathbf{Y}_{GC}^T \mathbf{Y}_{GC}}.$$

(6)

For imposing the weighting scheme, we introduced a weight tensor $\mathbf{W}$ of size $\mathbf{X}$ with non-negative weights. A higher weight results in a higher penalty on the error in fitting the entry, and thus prioritizing it. Given unfolded weighted matrices of $\mathbf{W}$ across genes, cell lines, and data types, $\mathbf{W}_G$, $\mathbf{W}_C$, and $\mathbf{W}_D$, the objective function and the update rules with the weighting scheme are (adapted from Blondel et al.[38]):

$$\min_{\mathbf{G} \geq 0} \sum_{i,j} \mathbf{W}_G(i,j) \left( \mathbf{Y}_{CD}\mathbf{G}^T(i,j) - \mathbf{X}_{(G)}(i,j) \right)^2,$$

(7)

$$\min_{\mathbf{C} \geq 0} \sum_{i,j} \mathbf{W}_C(i,j) \left( \mathbf{Y}_{GD}\mathbf{C}^T(i,j) - \mathbf{X}_{(C)}(i,j) \right)^2,$$

(8)

$$\min_{\mathbf{D} \geq 0} \sum_{i,j} \mathbf{W}_D(i,j) \left( \mathbf{Y}_{GC}\mathbf{D}^T(i,j) - \mathbf{X}_{(D)}(i,j) \right)^2,$$

(9)

$$\mathbf{G} = \mathbf{G} \odot \frac{\left( \mathbf{W}_G \odot \mathbf{X}_{(G)} \right)^T \mathbf{Y}_{CD}}{\left( \mathbf{W}_G^T \odot \left( \mathbf{G}\mathbf{Y}_{CD}^T \right) \right) \mathbf{Y}_{CD}},$$

(10)

$$\mathbf{C} = \mathbf{C} \odot \frac{\left( \mathbf{W}_C \odot \mathbf{X}_{(C)} \right)^T \mathbf{Y}_{GD}}{\left( \mathbf{W}_C^T \odot \left( \mathbf{C}\mathbf{Y}_{GD}^T \right) \right) \mathbf{Y}_{GD}},$$

(11)

$$\mathbf{D} = \mathbf{D} \odot \frac{\left( \mathbf{W}_D \odot \mathbf{X}_{(D)} \right)^T \mathbf{Y}_{GC}}{\left( \mathbf{W}_D^T \odot \left( \mathbf{D}\mathbf{Y}_{GC}^T \right) \right) \mathbf{Y}_{GC}},$$

(12)

where $(i,j)$ indicates the entry at $i$-th row and $j$-th column of the matrix. We obtained inverse of $||\mathbf{G}||_F^2$, $||\mathbf{C}||_F^2$, $||\mathbf{D}||_F^2$ (i.e. squared Frobenius norm) and used them for values in $\mathbf{W}_G$, $\mathbf{W}_C$, and $\mathbf{W}_D$, respectively, so that data types with less variance obtain high weights and are effectively integrated.

Orthogonality constraint is imposed only on one of the factor matrices, gene factor matrix

$$\mathbf{G}: \min_{\mathbf{G} \geq 0} \sum_{i,j} \mathbf{W}_G(i,j) \left( \mathbf{Y}_{CD}\mathbf{G}^T(i,j) - \mathbf{X}_{(G)}(i,j) \right)^2 \text{ s.t. } \mathbf{G}^T\mathbf{G} = \mathbf{I}$$

(13)

Following the orthogonal NMF developed by Yoo et al.[39], the multiplicative update rule of $G$ with orthogonality constraint can be formulated as the follows:

$$\Delta E = [\Delta E]^+ - [\Delta E]^- = \left( \mathbf{W}_G^T \odot \left( \mathbf{G}\mathbf{Y}_{CD}^T \right) \right) \mathbf{Y}_{CD} - \left( \mathbf{W}_G \odot \mathbf{X}_{(G)} \right)^T \mathbf{Y}_{CD},$$

(14)

$$\tilde{\Delta} E = \Delta E - \mathbf{G}(\Delta E)^T \mathbf{G}$$
$$= \left( \mathbf{W}_G^T \odot \left( \mathbf{G}\mathbf{Y}_{CD}^T \right) \right) \mathbf{Y}_{CD} - \left( \mathbf{W}_G \odot \mathbf{X}_{(G)} \right)^T \mathbf{Y}_{CD}$$
$$- \mathbf{G}\left( \left( \mathbf{W}_G^T \odot \left( \mathbf{G}\mathbf{Y}_{CD}^T \right) \right) \mathbf{Y}_{CD} - \left( \mathbf{W}_G \odot \mathbf{X}_{(G)} \right)^T \mathbf{Y}_{CD} \right)^T \mathbf{G}$$
$$= \left( \mathbf{W}_G^T \odot \left( \mathbf{G}\mathbf{Y}_{CD}^T \right) \right) \mathbf{Y}_{CD} - \left( \mathbf{W}_G \odot \mathbf{X}_{(G)} \right)^T \mathbf{Y}_{CD} - \mathbf{G}\left( \left( \mathbf{W}_G^T \odot \left( \mathbf{G}\mathbf{Y}_{CD}^T \right) \right) \mathbf{Y}_{CD} \right)^T \mathbf{G}$$
$$+ \mathbf{G}\left( \left( \mathbf{W}_G \odot \mathbf{X}_{(G)} \right)^T \mathbf{Y}_{CD} \right)^T \mathbf{G},$$

(15)

and we can simplify the latter two terms as follows:

$$\mathbf{G}\left( \left( \mathbf{W}_G^T \odot \left( \mathbf{G}\mathbf{Y}_{CD}^T \right) \right) \mathbf{Y}_{CD} \right)^T \mathbf{G} = \mathbf{G}\mathbf{Y}_{CD}^T \left( \mathbf{W}_G^T \odot \left( \mathbf{G}\mathbf{Y}_{CD}^T \right) \right)^T \mathbf{G} = \mathbf{G}\mathbf{Y}_{CD}^T \left( \mathbf{W}_G \odot \left( \mathbf{G}\mathbf{Y}_{CD}^T \right)^T \right) \mathbf{G}$$
$$= \mathbf{G}\mathbf{Y}_{CD}^T \left( \mathbf{W}_G \odot \left( \mathbf{Y}_{CD}\mathbf{G}^T \right) \right) \mathbf{G} = \left( \mathbf{W}_G^T \odot \left( \mathbf{G}\mathbf{Y}_{CD}^T \right) \right) \mathbf{Y}_{CD}\mathbf{G}^T \mathbf{G} = \left( \mathbf{W}_G^T \odot \left( \mathbf{G}\mathbf{Y}_{CD}^T \right) \right) \mathbf{Y}_{CD},$$

(16)

$$\mathbf{G}\left( \left( \mathbf{W}_G \odot \mathbf{X}_{(G)} \right)^T \mathbf{Y}_{CD} \right)^T \mathbf{G} = \mathbf{G}\mathbf{Y}_{CD}^T \left( \mathbf{W}_G \odot \mathbf{X}_{(G)} \right) \mathbf{G} = \left( \mathbf{W}_G^T \odot \left( \mathbf{G}\mathbf{Y}_{CD}^T \right) \right) \mathbf{X}_{(G)}\mathbf{G}$$

(17)

since $\mathbf{G}^T\mathbf{G} = \mathbf{I}$ in Stiefel manifold[39]. Note that $\mathbf{G}\mathbf{Y}_{CD}^T \left( \mathbf{W}_G \odot \left( \mathbf{Y}_{CD}\mathbf{G}^T \right) \right) = \left( \mathbf{W}_G^T \odot \mathbf{G}\mathbf{Y}_{CD}^T \right) \mathbf{Y}_{CD}\mathbf{G}^T$ and $\mathbf{G}\mathbf{Y}_{CD}^T \left( \mathbf{W}_G \odot \mathbf{X}_{(G)} \right) \mathbf{G} = \left( \mathbf{W}_G^T \odot \left( \mathbf{G}\mathbf{Y}_{CD}^T \right) \right) \mathbf{X}_{(G)}\mathbf{G}$ hold

since the element-wise multiplication can switch order ($\mathbf{A} \odot \mathbf{B} = \mathbf{B} \odot \mathbf{A}$; $\mathbf{A}(\mathbf{B} \odot \mathbf{C}) = (\mathbf{B}^T \odot \mathbf{A})\mathbf{C}$ when the size of A is the same as B), and size of $\mathbf{W}_G$ is the same as $\mathbf{Y}_{CD}\mathbf{G}^T$. By canceling the two identical terms, we have the gradient on the Stiefel manifold for the objective function:

$$\tilde{\Delta}\mathbf{E} = \Delta\mathbf{E} - \mathbf{G}(\Delta\mathbf{E})^T\mathbf{G} = \left(\mathbf{W}_G^T \odot (\mathbf{G}\mathbf{Y}_{CD}^T)\right)\mathbf{X}_{(G)}\mathbf{G} - \left(\mathbf{W}_G \odot \mathbf{X}_{(G)}\right)^T\mathbf{Y}_{CD}, \quad (18)$$

From which we can generate a multiplicative update rule as follows:

$$\mathbf{G} = \mathbf{G} \odot \frac{\left(\mathbf{W}_G \odot \mathbf{X}_{(G)}\right)^T\mathbf{Y}_{CD}}{\left(\mathbf{W}_G^T \odot (\mathbf{G}\mathbf{Y}_{CD}^T)\right)\mathbf{X}_{(G)}\mathbf{G}}. \quad (19)$$

Note that numerator of the above update rule remained unchanged regardless of the constraint. Orthogonal constraint strongly affects individual genes to be involved in multiple factors, and hence encourage capturing shared alteration patterns in different data types of the same gene. However, a gene can be involved in multiple biological processes captured by multiple factors, and thus the strict orthogonal constraint can significantly increase the reconstruction error. We introduced a tuning parameter $w$ to control the strength of the constraint, which is implemented as the mixing coefficients of the two objective functions (with and without the orthogonal constraint). Similarly, to obtain the update rule, we calculated the decomposed gradient as below:

$$(1-w)\Delta\mathbf{E} + w\tilde{\Delta}\mathbf{E} = (1-w)\left(\mathbf{W}_G^T \odot (\mathbf{G}\mathbf{Y}_{CD}^T)\right)\mathbf{Y}_{CD}$$
$$+ w\left(\mathbf{W}_G^T \odot (\mathbf{G}\mathbf{Y}_{CD}^T)\right)\mathbf{X}_{(G)}\mathbf{G} - \left(\mathbf{W}_G \odot \mathbf{X}_{(G)}\right)^T\mathbf{Y}_{CD}, \quad (20)$$

from which the final objective function is derived:

$$\mathbf{G} = \mathbf{G} \odot \frac{\left(\mathbf{W}_G \odot \mathbf{X}_{(G)}\right)^T\mathbf{Y}_{CD}}{(1-w)\left(\mathbf{W}_G^T \odot (\mathbf{G}\mathbf{Y}_{CD}^T)\right)\mathbf{Y}_{CD} + w\left(\mathbf{W}_G^T \odot (\mathbf{G}\mathbf{Y}_{CD}^T)\right)\mathbf{X}_{(G)}\mathbf{G}}. \quad (21)$$

The weight parameter $w$, which spans 0 to 1, serves as a soft selection variable between denominators of update rule of G with ($w = 1$) and without ($w = 0$) the orthogonality constraint. We set $w = 0.1$ after testing ranges of weights since the reconstruction error achieved by WON-PARAFAC starts to increase after $w = 0.1$ (Supplementary Fig. 2). The estimated factors are normalized and sorted based on l2-norms. Specifically, the multiplication of l2-norms for gene-factors, cell-factor and DT-factor are first calculated, followed by sorting the factors in decreasing order of the values.

**Selecting the number of factors.** Selecting the number of factors, or selecting the number of factors in WON-PARAFAC, is done using: (1) Akaike information criterion (AIC)[40]; and (2) average cosine similarity between all pairs of factors. AIC is a measure based on information theory, which deals with the trade-off between the goodness of fit and complexity of a given model. Cosine similarity between factors measures how redundant factors are from the other factors on average. We calculated this measure on the Khatri-Rao product of gene and data type factor matrices, $\mathbf{Y}_{GD}$, to focus on dependency between captured alterations. We trained WON-PARAFAC model with a sequence of number of factors ranges from 10 to 200 with step size of 10 and assessed the aforementioned measurements. For each WON-PARAFAC, absolute values of the leading eigenvectors are used as initial factors. Among the models, model with low AIC and cosine similarity was chosen.

**Cell set enrichment analysis.** For associating cell line factors (cell-factor) to tissue type, we assessed the enrichment of tissue types among the cell lines with high factor loadings. We adapted a simplified algorithm for gene set enrichment analysis[19]. Given a cell-factor contains non-negative weights $\mathbf{c}_i$ across cell lines and tissue type information of them, the enrichment score of the tissue is: $\sqrt{n}\frac{\sum_i(c_i - m)}{n\sigma}$, where $n$, $m$, and $\sigma$ are length, mean and standard deviation of the factor weights, respectively. The enrichment score is assumed to be normal random variable ($\sim N(0, 1)$), on which we perform right-tailed test to obtain $p$-values, followed by false discovery rate (FDR) correction. Tissue types with FDR < 0.2 are interpreted as significantly associated with the cell-factor.

**Gene set enrichment analysis.** For an unbiased functional interpretation, we first associated 130 factors and whole gene expression data (in total 16,244 genes) with linear regression analysis. This resulted in a 16,244 by 130 coefficient matrix, as each factor was regressed against every gene. Based on each column (factors) of the coefficient matrix, we tested for enriched gene sets using the enrichment scores as in the original GSEA[41]. We obtained null-hypothesis distribution by repeating the aforementioned procedures a 1000 times with randomly permuting samples in the gene expression matrix. The FDR is calculated from the null-hypothesis distribution. Gene sets with FDR < 0.2 are claimed to be significantly enriched. The R package is available online at https://doi.org/10.5281/zenodo.438018.

**Elastic net regression model for predicting drug responses.** We trained an elastic net (EN) regression model using glmnet R package[42]. For each of the 265 compounds, two EN regression models are constructed using either raw molecular

features or cell-factor loadings. For each regression model, 10-fold cross-validation was performed after which regularization penalty parameter lambda with the minimum loss is selected. To assess prediction performance of compounds assessed for cell lines in an unbiased manner, a nested double-loop cross-validation was performed. Partitioning of data kept the same across all of the models for fair comparisons.

**Construction of network for combined interpretation.** Given associations of 130 factors with tissue types, gene sets, and drug responses, we constructed networks for combined functional interpretations. We used visNetwork[43], and igraph[44] R packages for visualization. For ERBB2 example, we selected all factors associated with BIBW2992 response and top-5 gene sets associated with the factors (top three KEGG, top BP and Hallmark terms from MSigDB). For TOP1 poisons, we took the top six factors and top 4 gene sets associated with the factors (top KEGG and BP and top two Hallmark terms). For both networks, all the tissue types associated with at least one of the factors are included.

**Patient-derived xenograft data.** Processed gene expression, copy number, mutation, and drug response data of PDX encyclopedia is obtained for 399 PDXs with the molecular profiles available[4]. The gene expression data is standardized in the same way as cell line data by taking average and standard deviation of expression levels across pan-cancer PDX samples. Copy number and mutation data are converted to gain/loss/neutral (1, −1, 0) and wild-type/mutated (0, 1), respectively. For each data type, we constructed an 1815 gene by 399 PDX matrix while maintaining the order of genes and data types used in cell lines, followed by stacking them also in the same order to obtain a cube. Finally, WON-PARAFAC is applied to the cube and a permuted cube (i.e. stacked matrixes after permuting both samples and features) to obtain PDX factor matrix $\mathbf{P}$ (in place of $\mathbf{C}$) factor matrix $\mathbf{G}$ and data type matrix $\mathbf{D}$. Specifically, only the following objective function is optimized:

$$\min_{\mathbf{C} \geq 0}||\mathbf{Y}_{GD}\mathbf{P}^T - \mathbf{X}_{(P)}||_F^2. \quad (22)$$

The update rule of $\mathbf{P}$ is the same as $\mathbf{C}$ in WON-PARAFAC.

**Separation of two model systems in feature space.** For both raw features and factors, PDXs and cell lines are projected into two-dimensional space using $t$-distributed Stochastic Neighbor Embedding ($t$-SNE)[31]. In the projected space, distribution of PDXs and cell lines are compared per tissue type using Fisher criterion that measures within-group variance divided by between-group variance: $\frac{|\mathbf{m}_1 - \mathbf{m}_2|^2}{s_1^2 + s_2^2}$ where ($\mathbf{m}_1$, $\mathbf{m}_2$) and ($s_1^2$, $s_2^2$) are means and variances for the two groups, respectively. Due to the stochastic behavior of $t$-SNE, 100 random embedding are generated with random initialization, followed by measuring Fisher criterion between PDXs and cell lines per cancer type.

**Linking PDXs to tissue-types.** For the $i$th PDX a set of factors $T$ that is associated with a tissue type, we assessed the association of PDX to factors that are associated with the tissue type:

$$\frac{\sum_{\forall k \in T} \mathbf{P}(i, k)}{\sum_{\forall l} \mathbf{P}(i, l)}. \quad (23)$$

Note that the measure is close to 1 if variation in $i$th PDX is mostly captured by $T$, and close to 0 if they are completely independent.

**Reporting summary.** Further information on research design is available in the Nature Research Reporting Summary linked to this article.

## Data availability

Genomics data are downloaded from https://www.cancerrxgene.org for cell lines and from supplementary table of the PDX encyclopedia study[4] for PDXs. For the 130 identified factors and their associations to tissue type and drug responses in Supplementary Data 1; and comprehensive ShinyApp based on *visNetwork*[43], an R package for constructing interactive networks, is also available at https://ccb.nki.nl/software/won-parafac. The web-based tool offers a graphical user interface to explore all 130 factors, and to reconstruct some of the figures used in the paper.

## Code availability

WON-PARAFAC is implemented in MATLAB and the code is available at https://github.com/NKI-CCB/won-parafac. The WON-PARAFAC is based on and requires TENSOR TOOLBOX[45], an extensive MATLAB toolbox for handling tensors and tensor decomposition.

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

## Acknowledgements

This research was supported by an Alpe d'HuZes/KWF Bas Mulder Award and a VIDI grant to WZ and Alpe d'HuZes/STD(12725)/ERC-synergy grant to LFAW. We express our gratitude to our colleagues from the Computational Cancer Biology group and the Zwart Lab for useful discussions and constructive criticism. Nanne Aben is thanked for sharing great insights. We acknowledge the RHPC facility for providing resources. All authors are part of the Oncode Institute.

## Author contributions

Y.K., T.B., L.F.A.W., W.Z. and D.J.V. designed the experiments; Y.K. and T.B. performed the experiments, L.F.A.W., W.Z. and D.J.V. supervised the experiments.

## Competing interests

W.Z. reports funding to the institute from Astellas Pharma, L.F.A.W. reports funding to the institute from Genmab BV. Y.K., T.B. and D.J.V. report no competing interests.
