## [Peer Review File · Nature Communications]

Reviewers' comments:

Reviewer #1 (Remarks to the Author):

Summary

The authors propose a factorization model for three-way data which is formed by stacking (samples X genes) matrices from different data types (mutation, copy number, and gene expression data). The authors extend parallel factor analysis (PARAFAC) by adding the weighting scheme, a non-negativity constraint and orthogonality constraint to the factorization model. To ensure the non-negativity and orthogonality constraints, they derive multiplicative update rules for (gene, cell line and data type) factors in the model. Based on factors learned from data (GDSC or PDXE), the authors provide several analysis results that relate factors to tissue type, pathways, and treatment response.

This manuscript addresses very important problem with great interests to general audience from computational biology, bioinformatics, machine learning to cancer biologist. Integrating multiple data types and building predictive model based on the integrated data to identify predictive biomarker to predict treatment response is long standing challenging problem. The key idea of this manuscript is to deliver interpretable drug-gene association using factor analysis. The authors show that properly integrating multiple data types across cancer (pan-cancer analysis) could provide reliable biomarker (e.g., genes, pathways, etc.) that are associated with treatment response. The authors provide the experimental results showing that the proposed method achieved more accurate treatment prediction results in an independent PDX experiment.

The followings are my major comments:

Although, the proposed method provides a simple and compact way to compress high dimensional multiple data types, there are concerns regarding then current version of the manuscript/proposed method. I believe the correctness or proof of the proposed method could be provided and substantially improved with some of clarification and editing.

1) Technical correctness

- In line 391 (and 395), it seems that the weighted version of the subproblems would be given incorrectly. They should be defined as follows (Here is only the first case provided):

$$\min_{\{G\}} \sum_{i,j} [W_{\{G\}} \odot (Y_{\{CD\}} G^T - X_{\{G\}}) \odot (Y_{\{CD\}} G^T - X_{\{G\}})]$$

But, the multiplicative update rules the authors provide are derived from the corrected ones. I believe this would be simple error thus the authors should check out/correct the equations accordingly.

- Regarding the weighting scheme, it would be better to explicitly explain in the manuscript how the weights are set (not just citing a paper). This would provide more clear insight/view of the proposed method.

- In line 397, it seems that deriving the multiplicative rule with the orthogonality constraint is not straightforward. I tried to do it by myself based on the equation (27) in [Ref. 30, Yoo et al.], but found that the second and fourth terms are not canceled out because of the weights. I think it would be better if the authors provide detailed derivations in.

- In line 404, I think that the authors should provide the mathematical background for this update rule. The multiplicative update rule for NMF is derived from gradient descent, and the authors can show the convergence of methods based on multiplicative update rules (please see Lee and Seung, NIPS 2000). It is not clear what would be the corresponding objective function for the update rule when the parameter w is $0 < w < 1$? If the proposed update rule is derived from a different assumption rather than that of the standard NMF, the authors should prove the validity of their proposed algorithm (e.g., convergence analysis).

2) Comparison with two-way methods

It would be great if the authors could provide what additional information can be obtained from the proposed method (using all the data types) compared to a general NMF (using just single data

type such as gene expression data). Especially, In terms of predictive performance for treatment response, it will be great if the authors could provide some explanation whether integrating multiple types of data could give better/superior prediction performance compared to the use of single data type with simple NMF method.

Overall, the manuscript is well written and the experiments are properly done. The authors also provide source code and data to ensure to check reproducibility of the proposed method.

Reviewer #2 (Remarks to the Author):

The authors developed a computational method that can predict drug sensitivity from genomic datasets. It has novelty as they made improvement of conventional parallel factor analysis which provides better interpretation of biological features. This tool and source code are available at their webpage which could aid many potential scientist in the community. The performance of the tool was compared to other previously developed ones in its same-kind. The authors claim that the most interesting and advanced aspect of this tool is dimension reduction of high-dimension datasets and also comprehensive interpretation of biological features in given datasets. It could be useful and accelerate drug sensitivity research, however there are some minor issues. It is well-written and an easy-to-read paper, however there are also some issues that could be improved for the readers.

1. They need to do compare with some other data interaction methods. Such as
 - 1.1. Review papers about data integration methods (<https://www.ncbi.nlm.nih.gov/pmc/articles/PMC5472696/>)
 - 1.2. Matrix factorisation methods (iCluster)
 - 1.3. Network based methods (PARADIGM)
 - 1.4. Bayesian based methods
 - 1.5. So, they need to compare with those method about how their integration methods is good. For example, their method allows more precisely to predict drug sensitivity.
2. They need to justify why EN rather than other machine learning algorithms.
3. They need to add full name for the y-axis of Fig S2.
4. It would necessary to rephrase the lines 193-195.
5. They need more biological interpretation about Factors.
 - 5.1. How Factor are associated with some biological function?
 - 5.2. How Factor are associated with drug sensitivity? For example, genes, which factors are in, have same target genes
 - 5.3. They only show some cases. It looks like they only show some good cases.
6. There is an error in Figure 2G and (Page 6 line 162 to Page 7 line 166) "Factor 12 is associated with sensitivity to Afatinib(BIBW2992)". However, in Figure 2G, EN coefficient of BIBW2992 of Factor 12 is 0.
7. It would be helpful to see a figure that shows the association of these factors, similar to Fig 2E.
8. They need to do compare with some other drug sensitivity prediction method.
 - 8.1. Compare with other methods
 - 8.1.1. Raw feature vs Their method (Factor)
 - 8.1.2. Other methods that use PARAFAC
9. They need to more explanation about how they see Sensitivity or resistance
 - 9.1. For example, there is 4 factors are associated with a drug. 3 factors are positive, a factor is negative. Then is it sensitivity?
10. WON-PARAFAC provides very good interpretability of the biology in the given data at least compared to CNN for example. But it will be worth to discuss the advantage of this type of approach compared to other unsupervised/supervised machine learning algorithms in the Discussion.

11. Supp Table should have a cover page that explains details of each tab in the excel file.
12. The link for the code is broken. <http://ccb.nki.nl/software/wonparafac>

Reviewer #3 (Remarks to the Author):

The authors used a non-negative matrix factorization (NMF) method in conjunction with parallel factor analysis (PARAFAC) to enable connection between multi-layers molecular data to drug response measured in vitro. The idea of data integration with interpretable result is attractive. The method is relatively straightforward. However, one of the major conclusions regarding the factor-based method is outperforming the raw feature-based method is not well supported by the results presented. There are also some critical technical/biological issues need to be addressed.

Major issues:

1. In the background section, evidence on data integration outperforms single layer of information (eg gene expression) should be provided.
2. It is unclear how the 1815 cancer related genes were selected and more importantly the rationale for focusing on these small number of genes. The selection of genes to start with could change the results drastically and affect interpretation of the findings and conclusions. Therefore, an alternative gene selection method should be attempted and compared to the current findings.
3. The definition of GE +/- or CNV +/- is rather arbitrary. These criteria should be tested for robustness. With over 55 types of cancer cell lines, many exhibit tissue specific gene expression. Therefore for large number of genes, the expression is likely non-normally distributed, which make comparing to the mean an issue. Further, rationale should be provide for the choice of a greater than 5 copy define as CNV+ in cell line; since gain/loss/neutral (1, -1, 0) was used for PDX model.
4. Pg 6-7, results for interpretation of the factors. The authors listed the potential positive support for the findings, but did not mention how to interpret negative finding (false negative). For example, for factor 12, where when there is high ERBB2 expression, there is sensitivity to Afatinib. However, several other drugs that are also targeting ERBB2 (ie trastuzumab, lapatinib) and were not identified. Some explanation and potential interpretation of negative findings is needed.
5. Pg 7, on the global assessment of the factors, again, the interpretation is unclear. The authors picked the top 30 factors out of 130 total factors to say that they are enriched in commonly known cancer pathways. What about doing the top 10 factors, or all 130? How did this contribute to our current knowledge?
6. Pg 7, it is unclear what the section on "treatment response prediction based on the factors" is to show. Given the factors were derived from the 1815 genes in 3 data types, it is expected that the current model is highly correlated with models built using raw features. If anything this section demonstrated that within sample the current model perform worse than the models with raw features. Then why not build the model with raw features.
7. Pg 8, the authors use camptothecin, SN-38 as examples to demonstrate the biological relevancy of the current model. However, the current indication of these drugs is for colon cancer, where Fig 4D suggest colon cancer is one of the least sensitive cancer type to these drugs. These are not supportive of the text.
8. Results from method comparison conducted using limited number of PDX models on a handful of drugs were descriptive rather than definitive. They do not support the clear distinction of the superior performance of the factor methods.

Minor issue:

1. Unclear in the method p17, where the number 16,244 gene come from?
2. Pg 18, why only 399 PDX matrix?
3. Pg 8, line 214-216, the statement of "the cancer types strongly associated with ...head and neck lines..." need to show p values. Also, fig 4B need to show what others types are, since it appears

that afatinib sensitivity may be similar between head and neck cancer and some sub types of cancer included under others.

4. Pg 9, line 244-247, the statement of "among the cancer types, PDAC... were better reconstructed than BRCA...". This needs to be support by p value.

5. Pg 9, lin 254-257, "The cell-line/PDX (cell/pdx)-factors are distributed similarly in contrast to the raw feature representation, except for CRC...". Given the way the PDX factors were generated by fixed the cell line-derived gene-factor and Dt-factor, it is not surprising that the it performed better than raw features.

6. Fig 5F needs p values.

Response to comments

Reviewer #1 (Remarks to the Author):

Summary

The authors propose a factorization model for three-way data which is formed by stacking (samples X genes) matrices from different data types (mutation, copy number, and gene expression data). The authors extend parallel factor analysis (PARAFAC) by adding the weighting scheme, a non-negativity constraint and orthogonality constraint to the factorization model. To ensure the non-negativity and orthogonality constraints, they derive multiplicative update rules for (gene, cell line and data type) factors in the model. Based on factors learned from data (GDSC or PDXE), the authors provide several analysis results that relate factors to tissue type, pathways, and treatment response.

This manuscript addresses very important problem with great interests to general audience from computational biology, bioinformatics, machine learning to cancer biologist. Integrating multiple data types and building predictive model based on the integrated data to identify predictive biomarker to predict treatment response is long standing challenging problem. The key idea of this manuscript is to deliver interpretable drug-gene association using factor analysis. The authors show that properly integrating multiple data types across cancer (pan-cancer analysis) could provide reliable biomarker (e.g., genes, pathways, etc.) that are associated with treatment response. The authors provide the experimental results showing that the proposed method achieved more accurate treatment prediction results in an independent PDX experiment.

[Response]-----

We thank the reviewer for the constructive remarks and valuable feedback. We are delighted to hear the reviewer appreciates our work, and addresses a very important problem. We have addressed the reviewer comments as thoroughly as we could, which we believe these changes have substantially improved the quality of the manuscript. The reviewer comments are printed in regular font, our comments are provided in bold font.

The followings are my major comments:

Although, the proposed method provides a simple and compact way to compress high dimensional multiple data types, there are concerns regarding then current version of the manuscript/proposed method. I believe the correctness or proof of the proposed method could be provided and substantially improved with some of clarification and editing.

1) Technical correctness

- In line 391 (and 395), it seems that the weighted version of the subproblems would be given incorrectly. They should be defined as follows (Here is only the first case provided):

$$\min_{G \geq 0} \sum_{i,j} [W_{\{G\}} \odot (Y_{\{CD\}} G^{\wedge T} - X_{\{(G)\}}) \odot (Y_{\{CD\}} G^{\wedge T} - X_{\{(G)\}})]$$

But, the multiplicative update rules the authors provide are derived from the corrected ones. I believe this would be simple error thus the authors should check out/correct the equations accordingly.

[Response]-----

We are grateful to the reviewer for pointing out the error, which was indeed a mistake in the formulation. We do point out that the correct formula was used in the actual analyses. We corrected the error and double-checked if the same error appeared elsewhere.

Changes to the manuscript:

- Equations in “Weighted Orthogonal Non-negative PARAFAC (WON-PARAFAC)” of Materials and Methods

- Regarding the weighting scheme, it would be better to explicitly explain in the manuscript how the weights are set (not just citing a paper). This would provide more clear insight/view of the proposed method.

[Response]-----

We thank the reviewer for pointing out an oversight in the description of the introduction of the method. Based on the comment we now introduce the weighting scheme in “Weighted Orthogonal Non-negative PARAFAC (WON-PARAFAC)”, in the Methods section. The weighting scheme uses the inverse of the squared Frobenius norm of each data type matrix to balance their relative factor contributions.

We also added a new supplementary figure (Figure S4A) to show the variances across data types based on which weights are derived.

Changes to the manuscript:

- New supplementary Figure (Figure S4A)
- The last paragraph in “Weighted Orthogonal Non-negative PARAFAC (WON-PARAFAC)” of Materials and Methods

- In line 397, it seems that deriving the multiplicative rule with the orthogonality constraint is not straightforward. I tried to do it by myself based on the equation (27) in [Ref. 30, Yoo et al.], but found that the second and fourth terms are not canceled out because of the weights. I think it would be better if the authors provide detailed derivations in.

[Response]-----

We expanded our methods section to include a more detailed derivation of the update rules (see “Weighted Orthogonal Non-negative PARAFAC (WON-PARAFAC)”). The key point is that within the Stiefel manifold, we can cancel out $G^T G = I$ due to the orthogonality. To cancel it out, we need to change the order of the terms by taking advantage of the property of the Hadamard product, which allows us to change the order of the operations.

Changes to the manuscript:

- New supplementary Figure (Figure S4A)
 - Derivations in “Weighted Orthogonal Non-negative PARAFAC (WON-PARAFAC)” of Materials and Methods
-

- In line 404, I think that the authors should provide the mathematical background for this update rule. The multiplicative update rule for NMF is derived from gradient descent, and the authors can show the convergence of methods based on multiplicative update rules (please see Lee and Seung, NIPS 2000). It is not clear what would be the corresponding objective function for the update rule when the parameter w is $0 < w < 1$? If the proposed update rule is derived from a different assumption rather than that of the standard NMF, the authors should prove the validity of their proposed algorithm (e.g., convergence analysis).

[Response]-----

Our apologies for the brevity in the derivation of the update rules. The update rule is still derived from the multiplicative update rule from the standard NMF, where the positive and negative part of the gradient is used as numerator and denominator, respectively. We updated the text to clarify this in “Weighted Orthogonal Non-negative PARAFAC (WON-PARAFAC)”, in the Methods section.

Changes to the manuscript:

- Derivations in “Weighted Orthogonal Non-negative PARAFAC (WON-PARAFAC)” of Materials and Methods
-

2) Comparison with two-way methods

It would be great if the authors could provide what additional information can be obtained from the proposed method (using all the data types) compared to a general NMF (using just single data type such as gene expression data). Especially, In terms of predictive performance for treatment response, it will be great if the authors could provide some explanation whether integrating multiple types of data could give better/superior prediction performance compared to the use of single data type with simple NMF method.

[Response]-----

A key feature of our method, which sets it apart from 2D NMF on only a single data type, is the *simultaneous* integration of *different* data types across genes. This provides researchers with basic, interpretable factors that together recapitulate the cell line data. With just gene expression, one might be able to achieve good predictive power, but the interpretation is a real challenge. The benefit of integrating multiple data types in the supervised setting, especially regarding interpretability, was intuitively illustrated in TANDEM¹. In contrast to TANDEM’s deflationary

approach where different data types are integrated in a stepwise fashion, we here illustrate the *simultaneous* integration of the multiple data types in the unsupervised setting, i.e. without considering (any) response information. This approach balances prediction and interpretation, and this uniquely enables us to rejoin alterations at multiple levels shared across many cancer models. Approaches that predict well but cannot improve biological understanding through interpretation are in the end just black box models that either work or fail without a comprehensive understanding of why they do so.

One of the advantages of integration approaches is that single data types, such a mutation and copy number data, could be poor predictors on their own, but can contribute to prediction performance in an integrative setting. Data integration per se does not guarantee better predictive performance, but is more likely to be robust. Specifically, we illustrate that integrative classifiers show larger robustness as these transfer more readily from cell lines to PDX models (Section “Invariance of the factors using patient-derived xenograft data”, Figure 5E-G).

Changes to the manuscript:

We have rewritten parts of the Introduction section to makes these points clearer to the reader.

Overall, the manuscript is well written and the experiments are properly done. The authors also provide source code and data to ensure to check reproducibility of the proposed method.

[Response]-----

We appreciate the input and suggestions from the reviewer, and we are pleased to learn that the reviewer finds the paper well written and analyses well performed. We followed the open-science policy on sharing the source code of our work to make our study as reproducible as possible.

Reviewer #2 (Remarks to the Author):

The authors developed a computational method that can predict drug sensitivity from genomic datasets. It has novelty as they made improvement of conventional parallel factor analysis which provides better interpretation of biological features. This tool and source code are available at their webpage which could aid many potential scientist in the community. The performance of the tool was compared to other previously developed ones in its same-kind. The authors claim that the most interesting and advanced aspect of this tool is dimension reduction of high-dimension datasets and also comprehensive interpretation of biological features in given datasets. It could be useful and accelerate drug sensitivity research, however there are some minor issues. It is well-written and an easy-to-read paper, however there are also some issues that could be improved for the readers.

1. They need to do compare with some other data interaction methods. Such as

1.1. Review papers about data integration methods

(<https://www.ncbi.nlm.nih.gov/pmc/articles/PMC5472696/>)

1.2. Matrix factorisation methods (iCluster)

1.3. Network based methods (PARADIGM)

1.4. Bayesian based methods

1.5. So, they need to compare with those method about how their integration methods is good. For example, their method allows more precisely to predict drug sensitivity.

[Response]-----

We would like to thank the reviewer for the constructive comments and suggestions. We are pleased to read that the reviewer appreciates our work, and finds our novel approach useful to accelerate drug sensitivity research. We especially thank the reviewer for the constructive suggestion to place our approach in context, the suggested review was particularly useful.

We have updated our introduction to include a systematic comparison of WON-PARAFAC with other the methods from the reference suggested by the reviewer. In that reference none of the matrix factorization-based methods (the most relevant category of method to our approach) takes into account the relationship between the features for the same gene across different data types, as we do in our approach. iCluster is also a matrix-based method for integrating multiple data types, also without this property. Most of the matrix-based integration methods also do not provide interpretation of the factors. PARADIGM predicts the activity of an individual entity such as a pathway or network, and was not designed for summarizing multiple genomics data types.

We also added, as reviewer suggests, a performance comparison of WON-PARAFAC to TANDEM¹, a recently introduced approach for integrating multiple data types in the supervised setting. This comparison shows similar performance for TANDEM and WON-PARAFAC. In contrast to TANDEM that selects features in a sequential fashion, and EN that only selects raw features from a dominant data type, WON-PARAFAC performs *simultaneous* integration of all data types (especially mutation and copy number data) and provides an easily accessible framework that enables biological interpretation of each factor (Figure 4). WON-PARAFAC balances prediction and interpretation, and this uniquely enables us to rejoin alterations at multiple levels shared across many cancer models

Changes to the manuscript:

We revised the introduction to address this point. Also, we included one extra figure (Supplementary Figure 1) to illustrate the distinction between PARAFAC and other matrix-based integration approaches. See paragraph 2 in the revised Introduction. Importantly, we added the comparison with TANDEM (Section “Treatment response prediction based on the factors”, Figure S13).

2. They need to justify why EN rather than other machine learning algorithms.

[Response]-----

We have adapted the manuscript and now provide a justification for using Elastic Net (EN) in the first paragraph of the Section “Interpretation of factors”. Through our experience with the GDSC1000 data, we consistently found EN to be an efficient predictor, both in terms of runtime and predictive performance. We find that other methods, in particular SVMs or Random Forests, also perform well but do not perform categorically better (or worse) in terms of predictive performance. Independent groups using other data also support these observations². As EN can select a small number of factors to predict drug response, it also gives us a better opportunity to learn which factors are predictive for drug responses using the network-based visualizations.

3. They need to add full name for the y-axis of Fig S2.

[Response]-----

We have added the y-axis label in the Fig S2 (now Figure S3 in the revised manuscript).

4. It would necessary to rephrase the lines 193-195.

[Response]-----

This sentence must have been overlooked in our proofing process, we agree with the reviewer that it is convoluted and difficult to understand. We have extensively rewritten the paragraph, which now reads: The sentence at lines 193-195:

“For example, EN predicts response to Afatinib, a well-predictable drug that targets EGFR and ERBB2 (Figure 3A-B), there are four sensitivity factors with negative EN-coefficients, where cell lines with high loadings on these factors tend to be sensitive to Afatinib (Figure 3C).”

Was replaced with:

“For example, factor based EN predicts response to Afatinib (inhibits EGFR and ERBB2), using four cell line factors (Figure 3C). First, we note that the contribution of these factors to prediction correlate with the absolute EN coefficient (Figure 3C). Second, all these factors have negative EN coefficients implying that a high value of the factor corresponds to a low response activity area, i.e. sensitivity to the drug (Figure 3C, top). This is in agreement with that fact that Factor 40 is associated with EGFR expression (Figure 2B), which inhibited by Afatinib (Figure 2G).”

5. They need more biological interpretation about Factors.

5.1. How Factor are associated with some biological function?

5.2. How Factor are associated with drug sensitivity? For example, genes, which factors are in, have same target genes

5.3. They only show some cases. It looks like they only show some good cases.

[Response]-----

We attempted to provide a biological interpretation of the factors from three angles: 1) tissue type enrichment; 2) pathway/biological process enrichment; and 3) drug responses prediction. This is further illustrated in Figure 2A.

Re: 5.1, the factors are associated with biological function using enrichment analyses on the genes of a factor. For example, Section *Invariance of the factors using patient-derived xenograft data*, in which a factor shared (Factor 57) between CRC and PDAC is linked to oxidative phosphorylation. Complete list of associations between factors and biological processes are listed in revised Supplementary Data.

Re 5.2, the factors are directly associated with drug response by EN, all factors that have non-zero coefficient are said to be associated. We noticed that the explanation of how factors can be linked to drug responses was lacking in clarity. We have therefore expanded the description of how linking is obtained from the Elastic Net result (see second paragraph in Section “Treatment response prediction based on the factors”).

As there are many associations between 130 factors and 265 drugs/55 cancer types/thousands of biological functions, we had to pick a few examples (in Figures 2 and 4). That is the reason why we provide Supplementary Data and a companion software package (a shiny app provisionally hosted at <https://wonparafac.danielvis.nl/> during the review process). This allows users to explore 130 factors and associated drugs/cancer types/biological functions at their leisure. Readers are invited to choose their own examples beyond what is listed in the paper. Also, in the revised supplementary data, readers can find all associations of 130 factors to drug/cancer types/biological functions.

6. There is an error in Figure 2G and (Page 6 line 162 to Page 7 line 166) “Factor 12 is associated with sensitivity to Afatinib(BIBW2992)”. However, in Figure 2G, EN coefficient of BIBW2992 of Factor 12 is 0.

[Response]-----

The reviewer is correct in stating that the presentation is confusing, we inadvertently showed two independent instances of this compound. While the factor loading is non-zero, the scaling

differences give rise to confusion. The heatmap in Fig2G now shows the normalized coefficient, and both instances of Afatinib/bibw2992 are now placed next to each other, select Factor 12 and the agree on the direction (sensitive).

7. It would be helpful to see a figure that shows the association of these factors, similar to Fig2E.

[Response]-----

We can associate factors to 1) biological processes; 2) enriched tissue types and 3) drug response (as in Figure 2). We provided visualization using heatmaps (in Figure 2) and networks (Figure 4). We had to focus on a handful examples due to the large number of associations that were being captured. Therefore, we decided, instead, to provide the Shiny app (<https://wonparafac.danielvis.nl/>) and also the tables contain all associations tested (Supplementary Data) that allows readers to explore the associations based on their questions, which allows unlimited comparisons to be made by the reader. With the app, readers can filter factors based on many criteria such as genes or drugs of interest. With this, the app enables the reader to further explore their own questions beyond the limited set of examples provided in the manuscript, for the sheer reason of space limitations.

8. They need to do compare with some other drug sensitivity prediction method.

8.1. Compare with other methods

8.1.1. Raw feature vs Their method (Factor)

8.1.2. Other methods that use PARAFAC

[Response]-----

As per the reviewers' suggestion, we more clearly position WON-PARAFAC in the method landscape, in which a key differentiating factor is that WON-PARAFAC aims to identify features that are interpretable and maintain predictive response. We revised our manuscript to make this point clear (see second paragraph of revised Introduction).

We performed the second comparison the reviewer suggested (8.1.1 raw features vs factors) to support 1) predictable information is preserved in the factors (Figure 3A); and 2) superior translatability from in-vitro to in-vivo (Figure 5E-F). Finally, regarding point 8.1.2; we are not aware of any method that predicts drug response based on PARAFAC.

9. They need to more explanation about how they see Sensitivity or resistance

9.1. For example, there is 4 factors are associated with a drug. 3 factors are positive, a factor is negative. Then is it sensitivity?

[Response]-----

The reviewer interpreted the factor correctly; negative regression values imply sensitivity, because it results in lower AUCs as output of the regression model. Since the factor loadings are all non-negative, a factor can contribute to the prediction by addition or subtraction only. Hence, the regression coefficients obtained from the elastic net define whether a factor is associated to sensitivity (lower AUC) or resistance (higher AUC). For the set of factors selected by EN to perform the prediction, only the individual factors are classified as sensitive/resistant association, not the ‘model’ of all four factors.

We have rephrased the original sentence, as it was complicated to read (second paragraph of “Treatment response prediction based on the factors” in Result section).

10. WON-PARAFAC provides very good interpretability of the biology in the given data at least compared to CNN for example. But it will be worth to discuss the advantage of this type of approach compared to other unsupervised/supervised machine learning algorithms in the Discussion.

[Response]-----

As the reviewer suggested, we strengthened our discussion on how our approach compared to other methods (see Discussion, second paragraph). Along the same line, we also strengthened this point in the Introduction (second paragraph) and Supplementary Figure 1.

11. Supp Table should have a cover page that explains details of each tab in the excel file.

[Response]-----

We have added a table of contents to the cover page in the revised supplementary data as requested.

12. The link for the code is broken. <http://ccb.nki.nl/software/wonparafac>

[Response]-----

During the review process the methods website is hosted at wonparafac.danielvis.nl. It is our honest oversight to not include the source code there. This has been amended. The link is active now. Thanks for checking.

Reviewer #3 (Remarks to the Author):

The authors used a non-negative matrix factorization (NMF) method in conjunction with parallel factor analysis (PARAFAC) to enable connection between multi-layers molecular data to drug response measured in vitro. The idea of data integration with interpretable result is attractive. The method is relatively straightforward. However, one of the major conclusions regarding the factor-based method is outperforming the raw feature-based method is not well supported by the results presented. There are also some critical technical/biological issues need to be addressed.

[Response]-----

We thank the reviewer for the constructive and supportive input, and are happy to see the reviewer finds the approach attractive. Below are responses to the points raised by the reviewer.

Major issues:

1. In the background section, evidence on data integration outperforms single layer of information (eg gene expression) should be provided.

[Response]-----

It has been shown that naïvely integrating data types does not necessary improve predictive performance. More specifically, gene expression (GEX) as a single data type outperforms other data types in terms of predictive performance¹. However, since GEX is notoriously difficult to interpret, such models are not always preferred. A simple but elegant two-stage approach has been proposed (TANDEM)¹, which first regresses out other data types and only attempts to explain the remaining residual with gene expression. However, TANDEM is also essentially a feature selection method that disregards most of the molecular features that make up the underlining biology of drug responses.

Here we propose a different approach that also attempts to integrate mutation and copy number data with GEX. In contrast to TANDEM, we perform this decomposition independent of drug the response data (i.e. unsupervised) and show that it still contains the relevant information to accurately predict drug response. Along with its predictive ability, the joint decomposition across data types effectively integrates the data and facilitates the biological interpretation (see Section “Constraints enhance data integration and interpretation in WON-PARAFAC”, paragraph 2). However, none of the integrative approaches outperform GEX based predictors – they achieve comparable performance but greatly enhance interpretability as well as transferability to PDX models.

Changes to the manuscript:

In the revised manuscript, we positioned WON-PARAFAC among other data integration methods, and thoroughly compared them in terms of the concept employed in the approach (See

Introduction, second paragraph and Supplementary Figure 1). All of the methods we compared do handle multiple data types, but none of them are applied for predicting drug responses.

2. It is unclear how the 1815 cancer related genes were selected and more importantly the rationale for focusing on these small number of genes. The selection of genes to start with could change the results drastically and affect interpretation of the findings and conclusions. Therefore, an alternative gene selection method should be attempted and compared to the current findings.

[Response]-----

Our apologies for brevity the on discussion about the gene selection. We have now extended our description of the gene selection in the Method section. The gene set that was used was designed for a clinical diagnostic workflow. While the majority of panels cover fewer than 250 genes (Vis et al, Annals Oncology, 2017³), we chose the largest comprehensive cancer panel we could find (Center for Personalized Cancer Treatment (CPCT⁴); 1977 genes). We intersect these genes with the genes for which expression data were available, which left us with 1815 genes.

We see little merit in using small clinical gene sets, because they are less complete than our current comprehensive gene set. On the other hand, adding rarely mutated genes will contribute very little to the factor (see **Figure S14) and are therefore unlikely to benefit the interpretation or prediction. Of note, for the gene set enrichment analysis, we used all genes for which expression data were available.**

4. Pg 6-7, results for interpretation of the factors. The authors listed the potential positive support for the findings, but did not mention how to interpret negative finding (false negative). For example, for factor 12, where when there is high ERBB2 expression, there is sensitivity to Afatinib. However, several other drugs that are also targeting ERBB2 (ie trastuzumab, lapatinib) and were not identified. Some explanation and potential interpretation of negative findings is needed.

[Response]-----

The two drugs (trastuzumab and lapatinib) for which no ERBB2 associations were identified, as correctly pointed out by the reviewer, are not part of our data set. Trastuzumab is not a part of the GDSC1000 screening data from Iorio et al, and lapatinib was not included due to the low number of cells screened (<50%). However, we do agree that we should also point out the limitations of our analysis. The EN model trained on WON-PARAFAC features captured high gene expression of ERBB2 as a predictor of Afatinib response but not a mutation in EGFR, which is the clinically accepted biomarker for Afatinib response. In light of this point and the previous point, it can be useful to introduce a bias for biomarkers employed in clinical practice.

Changes to the manuscript:

We discussed this point in the revised manuscript (section “Linking biology to treatment response through Network”, first paragraph and Discussion, fourth paragraph).

5. Pg 7, on the global assessment of the factors, again, the interpretation is unclear. The authors picked the top 30 factors out of 130 total factors to say that they are enriched in commonly known cancer pathways. What about doing the top 10 factors, or all 130? How did this contribute to our current knowledge?

[Response]-----

We may have inadvertently suggested to the reader that we picked the top 30 factors. Instead, we performed enrichment analysis using all pathways in MsigDB across all 130 factors and only chose to show the top 30 most frequently enriched pathways among the >5000 gene sets for visualization (bar plot in Figure 12; all information available on Shiny app and Supplementary Data). What this analysis adds is that the factors mostly reflect cancer context, as the pathway analysis (GSEA) is done in an unbiased manner, using the full gene expression data set without selecting only the cancer-related genes. In the pathway analysis (based on GSEA), the associations between the 130 factors and the full gene expression data set are assessed. Based on this comment/question, we have rephrased “Gene Set Enrichment Analysis (GSEA)” in the Methods section and “Interpretation of the factors”, third paragraph, to improve the clarity of the text.

6. Pg 7, it is unclear what the section on “treatment response prediction based on the factors” is to show. Given the factors were derived from the 1815 genes in 3 data types, it is expected that the current model is highly correlated with models built using raw features. If anything this section demonstrated that within sample the current model perform worse than the models with raw features. Then why not build the model with raw features.

[Response]-----

The section aims to show two things; 1) the derived factors contain a predictive signal, 2) the factors allow translation to an in-vivo model system. While the reviewer is correct in stating that the same input features are used, the additional decomposition step performed to obtain the factors makes the predictors sufficiently different to require testing their predictive capacity. Which is what we did and show in Fig 3A.

The raw features outperform the factors (within model system predictions) only in a select set of cases where the known biomarker is a specific mutation in a single gene. The strongest argument against sticking with the raw features is that the performance of a predictor trained on cell lines and applied to PDXs is better using the factors than the raw features. We believe that we have shown that in Figure 5, factor-based EN models are more robust as they maintain predictive performance when transferred from cell lines to PDXs, compared to raw feature-based EN models which show a drop in performance. We hope this point is clearer now in the revised manuscript, where the relevant changes include:

- 1. Second paragraph in the Introduction, posing the issue of interpretation for handling high-dimensional data**

2. P-values in Figure 5F showing significant drop in performance for PDX application, only in raw feature-based models.

7. Pg 8, the authors use camptothecin, SN-38 as examples to demonstrate the biological relevance of the current model. However, the current indication of these drugs is for colon cancer, where Fig 4D suggest colon cancer is one of the least sensitive cancer type to these drugs. These are not supportive of the text.

[Response]-----

The reviewer is correct in that CRC is commonly treated with Camptothecin, but the reality is that most patients do not respond (in fact, 10-20 % respond clinically⁵). Withholding ineffective treatments (especially those with high toxicity) is an important theme within personalized medicine. The reviewer is also correct in stating that there are cancer types (e.g., lymphoblastic leukemia) that appear more sensitive, but these are not typically treated with Camptothecin but with other types of chemotherapeutics. This practice is maintained, in spite of trials that have indicated that these cancer types can achieve better clinical response rates than CRC (39% response in lymphoma⁶) which is supportive of our results.

8. Results from method comparison conducted using limited number of PDX models on a handful of drugs were descriptive rather than definitive. They do not support the clear distinction of the superior performance of the factor methods.

[Response]-----

We do agree that having more examples would have made our analysis more comprehensive. However, the PDX encyclopedia is the single largest data set to date comprising molecular and drug response profiles for PDX's. Consequently, this is the best that can be currently done to investigate the transfer of response predictors from the GDSC1000 cell line collection to PDX models. In our analyses, we observed a significant drop in performance of raw feature-based EN models when transferring from cell lines to PDX models ($p=0.0024$; Figure 5F), while the factor-based EN models retained their performance level showing no significant performance drop ($p=0.056$; Figure 5F). Also, we found more drugs with significantly correlated response prediction in PDXs for factor-based ENs compared to ENs derived from raw features (Figure 5E). However, as we are aware that the number of samples is limited, we have been careful not to over-state the importance of the results.

Minor issue:

1. Unclear in the method p17, where the number 16,244 gene come from?

[Response]-----

The gene expression data (microarray) covers 16,244 genes. Therefore, it is merely the entire list from the platform/array. Following this comment, we have amended the text and this is now clearly mentioned in the method section of the revised manuscript

2. Pg 18, why only 399 PDX matrix?

[Response]-----

It is a fair point. Among the all PDXs generated in PDXE, the authors provided genomics data for 399 samples, which is still the single-largest omics dataset of PDX models generated to date. The limitation of 399 models due to sample availability is now more clearly mentioned in the method section of the revised manuscript.

3. Pg 8, line 214-216, the statement of “the cancer types strongly associated with ...head and neck lines...” need to show p values. Also, fig 4B need to show what others types are, since it appears that afatinib sensitivity may be similar between head and neck cancer and some sub types of cancer included under others.

[Response]-----

We thank the reviewer for this thoughtful comment. We performed a cell set enrichment analysis (CSEA) and used FDR of 0.2 as the threshold (see “Linking biology to treatment response through networks”, first paragraph). The thresholds can be changed in the Shiny-app we provided for producing a custom network.

Regarding the Figure 4B, we only show those that are associated with the selected factors, all the others are combined under the label ‘others’. We did not further itemize the 16 tissue types linked to the $AUC < 0.6$ (the sensitive lines), of which 9 types occur only once. The selected cancer types in Figure 4B consist with high fraction of sensitive lines (e.g. 12 out of 29 lines – head and neck, 6 out of 29 lines – esophagus - and 5 out of 44 - breast), indicating our analysis prioritized tissue types enriched with sensitive lines.

Given the low frequency of tissue types, we elected not to break down the other tissue type to maintain clarity. Instead, we revised the first paragraph of “Linking biology to treatment response through networks” to indicate high number of sensitive lines among the cancer type selected by EN model, taking head and neck as the example.

4. Pg 9, line 244-247, the statement of “among the cancer types, PDAC... were better reconstructed than BRCA...”. This needs to be support by p value.

[Response]-----

The pvalues are now provided.

5. Pg 9, lin 254-257, “The cell-line/PDX (cell/pdx)-factors are distributed similarly in contrast to the raw feature representation, except for CRC...”. Given the way the PDX factors were generated by fixed the cell line-derived gene-factor and Dt-factor, it is not surprising that the it performed better than raw features.

[Response]-----

Projecting the PDX on the basis that was identified on cell lines by itself does not guarantee that the PDX samples distribute similarly. If the model systems were substantially different from one another, this analysis would have revealed that. In no way were the two model systems’ samples were constraint to be distributed similarly. Since the analysis, using fishers criterion is applied on the level of the t-SNE data, we do not bias this analysis towards factors. Furthermore, we note that no quantifiable improvement was observed for CRCs indicating that the higher concordance between the platforms is not expected for all cases.

We have moved the t-SNE plots to the supplemental materials for clarity.

6. Fig 5F needs p values.

[Response]-----

We have added p-values on Figure 5F, there is a significant ($p=0.0024$) performance drop of raw-EN in PDXs.

Reference

1. Aben, N., Vis, D. J., Michaut, M. & Wessels, L. F. TANDEM: a two-stage approach to maximize interpretability of drug response models based on multiple molecular data types. *Bioinformatics* **32**, i413–i420 (2016).
2. Jang, I. S., Neto, E. C., Guinney, J., Friend, S. H. & Margolin, A. A. SYSTEMATIC ASSESSMENT OF ANALYTICAL METHODS FOR DRUG SENSITIVITY PREDICTION FROM CANCER CELL LINE DATA. *Pac Symp Biocomput* 63–74 (2014).
3. Vis, D. J. *et al.* Towards a global cancer knowledge network: dissecting the current international cancer genomic sequencing landscape. *Ann. Oncol.* **28**, 1145–1151 (2017).
4. Hoogstraat, M. *et al.* Genomic and transcriptomic plasticity in treatment-naïve ovarian cancer. *Genome Res.* **24**, 200–211 (2014).
5. Cunningham, D., Maroun, J., Vanhoefler, U. & Cutsem, E. V. Optimizing the Use of Irinotecan in Colorectal Cancer. *The Oncologist* **6**, 17–23 (2001).
6. Ota, K. *et al.* [Late phase II clinical study of irinotecan hydrochloride (CPT-11) in the treatment of malignant lymphoma and acute leukemia. The CPT-11 Research Group for Hematological Malignancies]. *Gan To Kagaku Ryoho* **21**, 1047–1055 (1994).

REVIEWERS' COMMENTS:

Reviewer #1 (Remarks to the Author):

All concerns/comments I made are well addressed by the authors.

Reviewer #2 (Remarks to the Author):

Thank you for providing all the information. They are adequately addressed and resolved standing issues that came out from the review. I am happy to see this manuscript being published in Nature Comm.

Reviewer #3 (Remarks to the Author):

The revision has addressed all my concerns.